# Distributed Deep Learning In Open Collaborations

**Michael Diskin**$^{*♡†}$    **Alexey Bukhtiyarov**$^{*†♣}$    **Max Ryabinin**$^{*†♡}$

**Lucile Saulnier**$^{‡}$    **Quentin Lhoest**$^{‡}$    **Anton Sinitsin**$^{†♡}$    **Dmitry Popov**$^{†♡}$

**Dmitry Pyrkin**$^{♡}$    **Maxim Kashirin**$^{♡}$    **Alexander Borzunov**$^{†♡}$

**Albert Villanova del Moral**$^{‡}$    **Denis Mazur**$^{♣}$    **Ilia Kobelev**$^{†♣}$    **Yacine Jernite**$^{‡}$

**Thomas Wolf**$^{‡}$    **Gennady Pekhimenko**$^{◇♠}$

† Yandex, Russia
‡ Hugging Face, USA
♡ HSE University, Russia
♣ Moscow Institute of Physics and Technology, Russia
◇ University of Toronto, Canada
♠ Vector Institute, Canada

## Abstract

Modern deep learning applications require increasingly more compute to train state-of-the-art models. To address this demand, large corporations and institutions use dedicated High-Performance Computing clusters, whose construction and maintenance are both environmentally costly and well beyond the budget of most organizations. As a result, some research directions become the exclusive domain of a few large industrial and even fewer academic actors. To alleviate this disparity, smaller groups may pool their computational resources and run collaborative experiments that benefit all participants. This paradigm, known as grid- or volunteer computing, has seen successful applications in numerous scientific areas. However, using this approach for machine learning is difficult due to high latency, asymmetric bandwidth, and several challenges unique to volunteer computing. In this work, we carefully analyze these constraints and propose a novel algorithmic framework designed specifically for collaborative training. We demonstrate the effectiveness of our approach for SwAV and ALBERT pretraining in realistic conditions and achieve performance comparable to traditional setups at a fraction of the cost. Finally, we provide a detailed report of successful collaborative language model pretraining with 40 participants.

## 1  Introduction

The deep learning community is becoming increasingly more reliant on transfer learning. In computer vision, pretraining convolutional networks on large image collections such as ImageNet [1] is the de facto standard for a wide range of applications ranging from object detection [2] and semantic segmentation [3] to image classification [4] and even learning perceptual similarity [5]. A growing number of natural language processing systems capitalize on language models with billions of

---

$^{*}$Equal contribution. Correspondence to `mryabinin0@gmail.com`
Detailed author contributions are listed at the end of the work.

35th Conference on Neural Information Processing Systems (NeurIPS 2021).

parameters [6, 7, 8, 9, 10, 11] trained on vast unlabeled corpora. Similar trends have emerged in areas such as speech processing [12], reinforcement learning [13], and computational biology [14, 15].

Training these models is a notoriously time-consuming and challenging task: it often requires hundreds of high-end GPU servers [10, 16] and would take multiple years on a single device [17]. Most academic and independent researchers simply cannot afford to train state-of-the-art models from scratch, which slows down scientific progress and practical adoption of deep learning.

Historically, the deep learning community has addressed this problem via "model hubs" or "model zoos" — public repositories for pretrained model checkpoints [18, 19, 20, 21]. These repositories have played a significant role in the democratization of deep learning, allowing everyone to reap the benefits of large-scale training runs conducted by corporations and universities with sufficient resources. However, model hubs are limited to a narrow subset of datasets and tasks that match the interests of model creators. For instance, in natural language processing, it is often difficult to find up-to-date models for more than a handful of languages [22]. In turn, computer vision hubs rarely feature models trained on drawings, satellite images, 3D renders, microscopy, or any other data that does not resemble ImageNet. As a result, many researchers in these areas can only work on problems for which there are available pretrained models rather than the problems that most need solving.

However, there might be an alternative way to obtain pretrained models: to train these models *collaboratively*. This approach, known as volunteer (or grid) computing, allows many independent parties to combine their computational resources and collectively perform large-scale experiments [23, 24, 25]. The raw compute performance of such collaborations often exceeds that of the fastest supercomputers [26]; however, fully utilizing it can be challenging due to several reasons. First, devices that contribute to collaborative experiments can range from GPU servers and high-end workstations to consumer-grade computers and even smartphones [27]. Second, most of these devices use household internet connection with limited bandwidth and low reliability. Third, participants in such projects often donate their hardware part-time, joining and leaving the experiment at will.

While it is theoretically possible to train neural networks on this kind of infrastructure, modern distributed training strategies are only efficient in a narrow range of conditions. For instance, training with Ring All-Reduce [28] works well for identical servers but suffers significant performance penalties from network latency or bandwidth variation [29]. Another technique known as Parameter Server can handle heterogeneous devices at the cost of being less scalable [30]. Applying any of these strategies outside their preferred conditions may significantly reduce the training throughput [31], which makes them difficult to apply in the volatile infrastructure of volunteer computing. This issue is further complicated by the unique limitations of volunteer devices, such as network address translation (NAT), regional access restrictions, or variations in performance.

In this study, we carefully analyze the above challenges and come up with a practical solution for **D**istribute**d** **D**eep **L**earning in **O**pen **C**ollaborations (DeDLOC). DeDLOC is based on a novel algorithm that adapts to the available hardware in order to maximize the training throughput. Depending on the infrastructure, DeDLOC can recover parameter servers [30], All-Reduce SGD [32], decentralized SGD [33], BytePS [34], or an intermediate strategy that combines all of them. Using this algorithm, we propose a system for collaborative training designed to accommodate a large number of heterogeneous devices with uneven compute, bandwidth, reliability, and network capabilities.

The contributions of our work can be summarized as follows:

- We analyze the unique challenges of distributed training in open collaborations and propose a practical recipe for training in these conditions.

- We formulate a novel distributed training algorithm that interpolates between traditional strategies to directly maximize the training performance for the available hardware.

- We verify the effectiveness of the proposed algorithm and system design for unsupervised pretraining of ALBERT-Large and SwAV under realistic conditions.

- We run collaborative training with actual volunteers, achieving competitive results to models trained on hundreds of data center GPUs. We also report insights on the collaborator activity and share the codebase for running similar experiments in the future[2].

---

[2]Code and training configurations are available at `github.com/yandex-research/DeDLOC`

## 2 Related work

### 2.1 Distributed training

In this work, we focus on distributed data-parallel training, where each device runs forward and backward pass of the entire model on a subset of training examples. While there are many alternative techniques [35, 36, 37], data-parallel is still the most popular strategy. Even the model-parallel approaches for extremely large models rely on data parallelism at the top level [37, 16, 38].

Training on multiple nodes was first implemented with parameter server (PS) [30]. This training strategy relies on a dedicated node that stores model parameters and executes optimization steps using the gradients sent by workers. In turn, worker nodes iteratively download the latest version of model parameters from the server, compute gradients and submit them back to the PS. This strategy is easy to implement and use, but it has an unavoidable bottleneck: the entire system performance is limited by the network throughput of a single server. Since then, the scientific community proposed numerous extensions to PS that alleviate the bottleneck by reducing the communication load [39, 40, 41, 42, 43], introducing asynchronous updates [44, 45] or training with multiple servers [46, 34].

The issue of uneven communication load has also inspired the development and widespread adoption of another group of methods that rely on All-Reduce for gradient averaging [47, 48, 49]. All-Reduce is a family of collective operations that allow nodes to efficiently aggregate (e.g. average) their local vectors and distribute the result across all devices [28, 50, 51]. Unlike parameter servers, All-Reduce assigns equal roles to all devices, making it easier to scale to a large number of homogeneous workers.

The popularity of AR-SGD sparked many practical applications for different scenarios. One particularly relevant application is elastic training [52, 53], which allows the user to add or remove workers at any point without interrupting the training run. While this bears a lot of similarity with collaborative training, we have found that elastic training systems are designed around global state synchronization, which makes them highly dependent on the homogeneity of the workers and their network connectivity. The overall efficiency is bounded by the performance of the lowest-performing node; as a result, introducing even a single low-bandwidth participant to such systems reduces the training speed by orders of magnitude.

Seeking to avoid the need for synchronization and centralized orchestration, the research community has developed decentralized training algorithms. These algorithms can be broadly divided into two categories: directly passing updates between peers [54, 55] or running All-Reduce in small alternating groups [56, 29]. Compared to PS and All-Reduce, both categories provide a greater degree of fault tolerance but often require more steps to converge due to delayed updates [33, 29].

Most practical use cases of the above techniques take place in HPC or cloud conditions, but there is one notable exception. In Federated Learning, multiple parties train a shared model on decentralized privacy-sensitive data that cannot be shared between devices [57]. For that reason, federated learning algorithms prioritize data privacy over training efficiency, often leaving most of the compute resources unused [58, 59]. For a more detailed overview of Federated Learning, refer to Appendix A.

### 2.2 Volunteer Computing

Volunteer computing (VC) is a paradigm of distributed computing where people donate the idle time of their desktops, smartphones, and other personal devices to solve a computationally hard problem collectively. This approach has seen successful applications in bioinformatics, physics and other scientific areas [60, 61, 62, 23, 63, 64, 65].

In all these applications, volunteer computing allows researchers to access vast computational resources. In Folding@home, over 700,000 volunteers have collectively contributed 2.43 exaFLOPs of compute to COVID-19 research in April of 2020 [26]. Another project named BOINC (Berkeley Open Infrastructure for Network Computing) brings together 41.548 petaFLOPs from over 790,000 active computers as of 17 March 2020 [25]. Volunteer computing systems were also the first "supercomputers" to reach 1 petaFLOP and 1 exaFLOP barriers [26, 66]. These results became possible due to the contributions of a broad range of devices from high-end workstations to smartphones and even gaming consoles [67].

Unfortunately, this compute diversity is also the main limitation of VC. Any volunteer computing system should be able to run on a wide range of available hardware and maintain integrity even if

some participants disconnect. Furthermore, the resources available to a project can vary over time, as most volunteers are only sharing their hardware when it is unused. Finally, volunteer devices are interconnected with a shared high latency network at typical home internet connection speeds.

As a result, there were only a few successful attempts to apply volunteer computing to machine learning workloads. One such project is MLC@Home [68], which relies on volunteers to train many small independent models. This specific problem can be solved with no direct communication between participants. By contrast, distributed training of a single model requires significantly more communication and does not allow a natural way to "restart" failed jobs. When it comes to distributed training of neural networks, most volunteer computing projects rely on parameter server architectures [69, 70, 71]. As a result, these systems are bounded by the throughput of parameter servers and the memory available on the weakest GPU. The only notable exception is Learning@home [72], which uses expert parallelism to train larger models spanning multiple computers; however, this approach has only been tested in simulated conditions.

# 3 Distributed Deep Learning in Open Collaborations

There are two unsolved challenges that stand in the way of practical collaborative training. The first challenge is algorithmic: how to maintain optimal training performance with dynamically changing hardware and network conditions? Another major challenge is ensuring consistent training outcomes with inconsistent composition of participants. Thus, we organize this section around these two issues:

- Section 3.1 provides a general overview of DeDLOC and explains how it maintains consistency in a dynamic environment.
- In Section 3.2, we describe the generalized communication strategy that maximizes training throughput by adapting to the currently available devices.
- In Section 3.3, we address system design challenges, such as circumventing NAT and firewalls, training on large datasets and managing collaborator access.

## 3.1 Ensuring training consistency

Many state-of-the-art models, notably GANs [73] and Transformers [74], require a strict training regimen. Deviating from the recommended batch size or introducing stale gradients may significantly affect the training outcome [75, 76, 77]. Since in a collaborative setting one has little control over the devices that participate in the experiment, it is almost guaranteed that the specific hardware setup will vary between runs and even during a single run. Without special precautions, these runs may result in models with vastly different final accuracy.

To avoid this pitfall, DeDLOC follows synchronous data-parallel training with fixed hyperparameters regardless of the number of collaborators. In order to compensate for relatively slow communication, we adopt training with extremely large batches [78, 49, 75, 79, 10], which allows peers to communicate less frequently. This strategy also provides a natural way to deal with heterogeneous hardware [80]: each device accumulates gradients at its own pace until the collaboration reaches the target batch size. Once ready, the collaborators exchange their gradients and perform one optimizer step. Using synchronous updates makes DeDLOC mathematically equivalent to large-batch training on a regular HPC cluster; see Appendix G for a more detailed explanation. Figure 1 gives a high-level visual explanation of this algorithm.

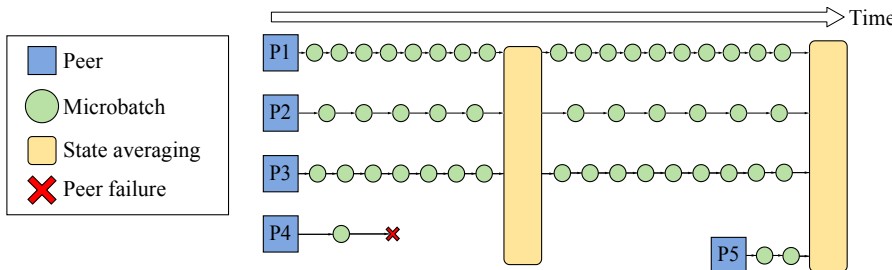

Figure 1: Two DeDLOC training iterations with example collaborator dynamics.

## 3.2 Adaptive averaging algorithm

As we discussed in Section 2.1, each distributed training algorithm has a narrow range of conditions where it can reach optimal performance. For instance, Ring All-Reduce works best on homogeneous hardware with low-latency communication, while Parameter Server strategy requires dedicated high-bandwidth devices that communicate with a large number of "workers". Since all devices are provided by volunteers, our training infrastructure is in a constant state of flux.

For instance, a collaboration can start with several homogeneous nodes that could be trained optimally with All-Reduce. If new participants bring devices with less bandwidth, it may be more efficient to use the original nodes as parameter servers. As more peers join, these servers will eventually become unable to handle the network load and the collaboration will need to switch to a different strategy.

Running efficient training on this kind of infrastructure requires a protocol that can dynamically assign roles to every peer given their hardware and network capabilities:

- **Compute performance:** Each peer $i \in 1, \ldots, n$ can compute gradients over $s_i$ samples per second. A peer that is unable to compute gradients (i.e. that has no GPU) will have $s_i{=}0$.
- **Bandwidth:** Peers communicate with a limited throughput: $d_i$ for download and $u_i$ for upload.
- **Geographical limitations:** In addition to individual bandwidth, the communication throughput between two peers $i, j$ is also restricted by $t_{ij}$ and $t_{ji}$ in each direction.

Given these constraints, our objective is to find a communication strategy that has the highest training throughput, that is, the one that *makes the most SGD steps with a target batch size $B$ per unit of time*. In turn, the training throughput of a collaboration depends on how we split the load among the participants. Each peer can be assigned to compute gradients over a subset of training examples, aggregate a part of those gradients from all peers, or both.

For simplicity and efficiency, we use delayed parameter updates (DPU) [81] — a technique that allows gradient computation and communication to run in parallel, at the cost of exactly one round of staleness. This strategy can improve time to convergence for a wide range of models, including Transformers [81, 82]. That said, our approach can be easily adapted to non-concurrent updates.

With DPU, the frequency of training updates is determined by either the time to compute gradients or the time to aggregate them, whichever takes longer. In total, a collaboration processes $\sum_{i=1}^{n} s_i \cdot c_i$ samples per second, where $c_i$ is the binary indicator denoting whether $i$-th peer is assigned to contribute gradients. Assuming the target batch size $B$, the frequency of the computation phase can be expressed as $F_{compute} = \sum_{i=1}^{n} s_i \cdot c_i / B$.

During the communication phase, each peer is first assigned to accumulate gradients over a fraction of model parameters. After that, everyone partitions their local gradients and sends each partition to the corresponding peer. On the other end, receiver nodes accumulate the gradients from all senders and return the average. In modern distributed training systems, this procedure is highly parallelized [34, 83]: a reducer can aggregate one chunk of gradients while downloading the next chunk and distributing the previous one back to the same senders.

In order to properly optimize the training throughput, we must account for this parallelism. As such, we explicitly define the speed $a_{ij}$ at which peer $i$ sends gradients to peer $j$ for aggregation. In turn, $j$-th peer aggregates gradients from all peers at the rate of the slowest sender $a_j = \min_{i:c_i=1} a_{ij}$. The senders can then get the aggregated results from the $j$-th reducer at $g_{ji} \leq a_j$. Finally, the total $a_{ij}$ and $g_{ij}$ for each peer cannot exceed their maximum download/upload speed. The only exception is that transfer within one node ($a_{ii}$, $g_{ii}$) does not count towards network throughput.

The frequency of the gradient aggregation phase is simply the rate at which the slowest peer can aggregate the full gradient vector: $F_{agg} = \min_i \sum_j g_{ji} / P$ , where $P$ is the number of model parameters. The final optimization problem can be formulated as follows:

$$
\begin{aligned}
\max_{a,g,c} \quad & \min \left( \frac{\sum_{i=1}^{n} s_i \cdot c_i}{B}, \ \frac{\min_i \sum_j g_{ji}}{P} \right) \\
\text{s.t.} \quad & g_{ij} \leq \min_{k:c_k=1} a_{ki} && \forall i, j \\
& \sum_{j \neq i} (a_{ji} + g_{ji}) \leq d_i && \forall i \\
& \sum_{j \neq i} (a_{ij} + g_{ij}) \leq u_i && \forall i \\
& a_{ij} + g_{ij} \leq t_{ij} && \forall i, j
\end{aligned}
\tag{1}
$$

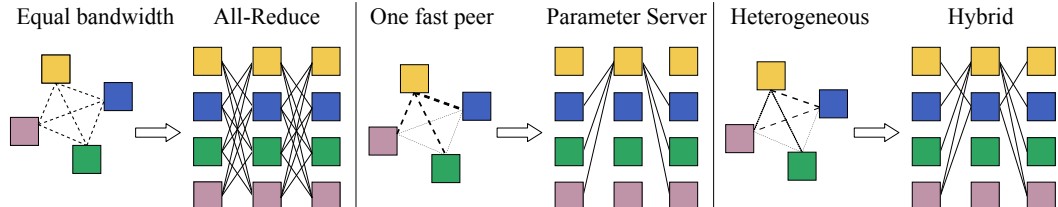

Figure 2: Example collaboration setups and corresponding strategies for optimal averaging. Each square represents one of the peers, line thickness denotes pairwise connection speed.

This problem must be solved regularly as participants are joining and leaving. Thus, we must ensure that the benefits of the optimal strategy outweigh the overhead of computing it. For that reason, we formulate optimal strategy search as a linear program that can be solved efficiently[3]. A more formal definition of problem (1) with detailed LP reduction can be found in Appendix B.

After this problem is solved, we assign each peer to aggregate a fraction of gradients proportional to $\min_j g_{ji}$. Peers with $c_i=1$ are also tasked with computing the gradients, while peers with $c_i=0$ remain idle and only participate in communication. This results in a natural division of labor. In the presence of many compute-heavy peers, some participants without accelerators will dedicate all their bandwidth to gradient aggregation instead of sending their local gradients.

**Node failures.** The resulting procedure can find the optimal communication strategy for averaging gradients across all participants. However, as the number of participants grows, it might be impractical to compute the global average due to node failures. Based on our experiments with several hundred active volunteers, most training iterations will have at least one participant with network issues. This implies that without necessary precautions, the entire averaging round will fail more often than it will succeed. To combat this issue, we use techniques [56, 29] that replace global averaging with several consecutive iterations in alternating groups of size $m$. The groups are chosen in such a way that the collaboration can obtain the exact average in $\log_m n$ steps. Furthermore, if any single participant fails, it will only affect his immediate group rather than the entire collaboration.

We adaptively choose the optimal group size $m$ based on the number of peers and their failure rates. This optimization problem is independent of Equation (1) and aims to maximize the rate at which collaborators can compute the global average. We elaborate on this procedure in Appendix C.

**Comparison with existing techniques.** Our method was designed as a generalization of existing data-parallel strategies that recovers them in special cases. To illustrate this idea, we provide example configurations for which DeDLOC recovers specific well-known strategies:

1. **AR-SGD:** a homogeneous collaboration with reliable peers will use Butterfly All-Reduce [84];
2. **Parameter Server:** adding a single participant with a very high bandwidth and low compute performance will turn the previous collaboration into a parameter server [30];
3. **BytePS:** participants with the same bandwidth as AR-SGD nodes, but without compute accelerators, will behave as auxiliary summation services from BytePS [34];
4. **Decentralized SGD:** any collaboration with a sufficiently high failure rate will converge to $m=2$. In this mode, all communication is performed between pairs of nodes, similarly to D-PSGD [33].

However, when training with actual volunteer devices, DeDLOC typically follows a hybrid communication scheme that differs from each of the above options. We display several examples of schemes that can arise as a solution for the optimal strategy search problem in Figure 2.

### 3.3   System design

Training with volunteer hardware requires specialized system architecture that can dynamically scale with collaboration size and recover from node failures. DeDLOC achieves these properties by operating as a swarm, similarly in spirit to BitTorrent [85] and I2P [86]. Individual peers coordinate by forming a Distributed Hash Table — a fully decentralized fault-tolerant key-value storage [87, 88]. Collaborators use this shared "dictionary" to count the number of accumulated gradients, find groups for averaging and keep track of the training progress.

---

[3]In our experiments, the LP solver consistently converged in $< 50$ms and was called $\approx 2$ times per minute.

DeDLOC ensures that all peers use up-to-date parameters by tracking the number of global steps of each peer. If a peer skips a step, it will observe that others made more steps and download the latest parameters and optimizer statistics from one of the up-to-date peers before resuming training.

In order to ensure the integrity of DHT throughout the training run, DeDLOC requires a few peers with stable internet access. These "backbone" peers are responsible for welcoming new collaborators and performing auxiliary functions, such as storing checkpoints and tracking learning curves. The only requirement for those peers is that at least one of them is available at all times. As such, the backbone peers can be hosted on inexpensive servers without GPU (see Appendix F for cost analysis).

All other devices are treated as regular collaborators. Depending on their hardware and network bandwidth, these devices can be assigned to (i) compute gradients, (ii) aggregate gradients computed by other peers or (iii) do both, according to the adaptive averaging algorithm. However, performing these steps with actual volunteer devices requires solving another set of challenges described below.

**Training under NAT and firewalls.** In addition to having uneven compute and network capabilities, volunteer devices also deviate from traditional servers in network configuration. One major difference is the use of Network Address Translation (NAT) [89] — the technology that allows multiple devices to share the same IP address. In practice, the majority of household and organizational computers around the world use one or multiple layers of NAT (see Appendix D for more details). Unfortunately for distributed training, NAT makes it harder to establish peer-to-peer connections [90].

When operating under NAT, DeDLOC participants use one of the following techniques:

1. **Hole punching:** use a third peer to temporarily open access to both devices. Once both peers are accessible, they can establish a direct connection and transfer data as usual [90];
2. **Circuit relays:** both devices connect to a relay (another peer that is mutually accessible), then forward all communication through that relay [91];
3. **Client mode:** if everything else fails, a peer can still send gradients to others without the need for incoming connections. This imposes an additional constraint $a_i = 0$ for Equation (1).

A similar set of strategies can be found in a wide range of distributed systems that rely on peer-to-peer communication, such as WebRTC, VoIP (IP telephony), and BitTorrent. Most of these systems rely on dedicated servers to establish connections between peers. However, in our case it is more appealing to use a fully decentralized NAT traversal where the regular peers perform hole punching and relaying by themselves. We describe this approach in more detail in Appendix E.

**Training on large datasets.** Many prospective applications of DeDLOC require training on large datasets that can take multiple hours to download. We circumvent this problem by allowing participants to download the data progressively during training. To support this behavior, we split the dataset into shards; upon joining the collaboration, a peer begins downloading examples shard by shard in a streaming fashion. Once the first several examples are obtained, a collaborator can begin training right away while downloading the rest of data in background.

To ensure that the training examples are independent and identically distributed, each participant loads shards in a different random order and uses a buffer to shuffle the data within each shard. Each participant loads the first $S = 10,000$ examples into a buffer, then randomly picks a training batch from this buffer and replaces the chosen examples with newly downloaded ones. In our experiments, we stream the training data from a dedicated storage service. However, this service can be replaced with a peer-to-peer data sharing protocol akin to BitTorrent; see Appendix H for details.

**Collaborator authentication.** Many prospective applications of DeDLOC need a way to keep track of individual peer contributions and protect against malicious peers. In our experiments, we achieve this using an allowlist authentication system that we describe in Appendix I.5.

## 4 Experiments

In this section, we evaluate the performance of DeDLOC in realistic collaborative training conditions. Our primary focus is on training models that are useful for a wide range of downstream tasks and thus would attract a large number of collaborators. One area that fits this description is self-supervised learning, i.e., learning reusable feature representations on large unlabeled datasets. First, we conduct controlled experiments on two popular self-supervised learning tasks in Sections 4.1 and 4.2. Then, we set up a real-world collaborative training run with volunteers and report our findings in Section 4.3.

## 4.1 Self-supervised learning of visual representations

Our first set of experiments uses SwAV [92] — a self-supervised learning technique that learns image representations by contrasting cluster assignments. Similarly to the original paper, we train the ResNet-50 [93] model on the ImageNet dataset [1] without labels. Our experiments follow the recommended training configuration [92, 94]: 2+6 random crops, early prototype freezing and a queue with 3,840 samples for each worker, LARS [78] optimizer, and 32,768 samples per batch across all workers. In this and further experiments, we use Hivemind [95] to implement the infrastructure for decentralized averaging. We train with three hardware setups: SERVER, WORKSTATION and HYBRID. The SERVER setup contains 8 workers, each with a single V100 GPU and 1 Gb/s symmetric bandwidth. In turn, the WORKSTATION setup consists of 16 nodes with 1080 Ti and 200 Mb/s bandwidth per worker. Finally, the HYBRID setup combines both previous configurations for a total of 24 nodes. Unlike servers, workstation GPUs train in full precision because they do not support accelerated float16 computations [96].

We report learning curves for each hardware configuration in Figure 3. As expected, the HYBRID setup converges the fastest, beating SERVER and WORKSTATION setups by 40% and 52% accordingly. When used in a supervised setting (Section 4.1 from the original paper), the model learned in this setup achieves a comparable accuracy of 72.2%. Another important observation is that the workstation-only experiment achieves reasonable training throughput despite using dated hardware. To provide more insight into the performance of DeDLOC, we also measure the time it takes to run averaging in different configurations. We report the mean over 100 averaging rounds; the standard deviation was below 1% in all setups. As demonstrated in Table 1, adaptive averaging does not affect the performance for homogeneous setups while running 1.9 times faster on the hybrid infrastructure.

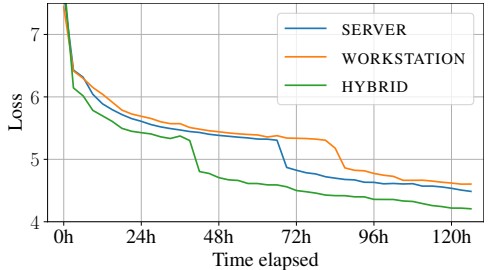

| Setup | Algorithm | | |
|---|---|---|---|
| | AR | PS | Ours |
| A: 8x1Gb/s | **1.19** | 4.73 | 1.20 |
| B: 16x0.2Gb/s | **5.3** | 39.6 | **5.3** |
| C: A + B | 5.69 | 14.1 | **2.96** |
| D: B + 1x2.5Gb/s | 5.3 | 3.22 | **3.18** |

Figure 3: SwAV pretraining performance.          Table 1: ResNet-50 averaging performance.

## 4.2 Self-supervised pretraining for language understanding

Next, we investigate how collaborative training performs for more complex models. In this experiment, we pretrain the ALBERT-large [7] masked language model on the WikiText-103 dataset [97]. We chose this setup for two reasons: first, ALBERT is very sensitive to the choice of hyperparameters, and specifically batch size, even more than regular Transformers [75]. This makes it easier to verify that DeDLOC can reproduce the training conditions of regular data-parallel training. Second, because of weight sharing, training ALBERT is relatively more compute- and less communication-intensive than regular BERT [6], which makes it possible to train with lower bandwidth.

As before, we follow the exact training configuration from the original paper, but use GPUs instead of TPUs. We use the implementation of ALBERT from the `transformers` library [99]. We run all experiments on cloud instances with Tesla T4 GPUs and report the training loss as a function of time, similarly to [17, 38]. In order to evaluate how DeDLOC performs with different network speeds, we consider the following setups on the same platform with controlled conditions:

- **High-bandwidth:** 16 workers, each with Tesla T4 and 25 Gb/s symmetric bandwidth;

- **Heterogeneous:** same, but with 4x 200 Mb/s, 8x 100 Mb/s and 4x 50 Mb/s bandwidths;

- **Heterogeneous + load balancing:** like Heterogeneous, but with adaptive averaging (Section 3.2);

- **Auxiliary peers:** the previous setup with 4 additional CPU-only peers at 1 Gb/s bandwidth.

- **Time-varying:** same as previous, but with 8 additional peers at 100 Mb/s. The extra peers are training part-time, jointly alternating between 8 hours of training and 8 hours of downtime.

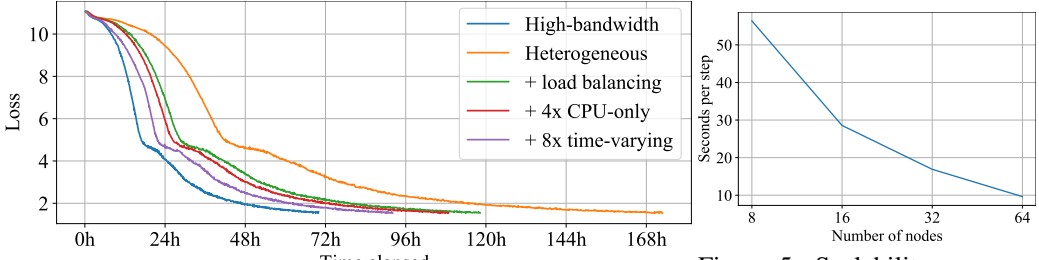

Figure 4: ALBERT pretraining performance.

Figure 5: Scalability measurements for ALBERT pretraining.

As one can see in Figure 4, naïve training with low-bandwidth peers results in an $\approx 2.5$x slowdown compared to high-bandwidth ones. Enabling load balancing accelerates that setup by $\approx 47\%$. This effect grows to over 60% when adding 4 auxiliary peers. Finally, adding 8 part-time peers allows the collaboration to train at 74% the speed of the high-bandwidth setup without sacrificing the training stability. This turns the latter setup into a viable alternative to traditional distributed training without the need for expensive infrastructure (see the cost analysis in Appendix F). In addition, we demonstrate the high scalability of DeDLOC in Figure 5, which was obtained by running the same experiment with a varying number of nodes and measuring the time between gradient descent steps.

### 4.3 Real-world collaborative training

For our final evaluation, we organized an actual collaborative training run with volunteer participants, who were asked to pretrain a Transformer masked language model for the Bengali language. This task was chosen deliberately to show the benefits of collaborative training: Bengali has over 230M native speakers who can benefit from recent advances in NLP, but there are few pretrained models available for this language. We recruited 30 Bengali-speaking volunteers and 10 outside collaborators. All participants received instructions for contributing with free cloud platforms and access to the code for training on local computers. To avoid bias, we did not encourage any specific form of participation: volunteers were free to choose what hardware they contributed and for how long.

Specifically, we trained the ALBERT-large model on Wikipedia and the Bengali part of the OSCAR [100] multilingual corpus. The model was named sahajBERT after conducting a poll among the participants. We adapted our preprocessing by following the best practices for the Bengali language described in Appendix I.3. To stream from a mix of Wikipedia and OSCAR, the training process iteratively sampled examples from one or the other dataset, as described in Section 3.3. We accounted for uneven size and quality of data by oversampling Wikipedia by a factor of 2, which resulted in mixing probabilities of 0.23 for Wikipedia and 0.77 for OSCAR. Other hyperparameters were set to the same values as in Section 4.2. Also, in Appendix I.7 we report the results of sahajBERT-XL — a four times larger model with a specialized architecture that used both GPU and TPU resources.

In total, the 40 volunteers contributed compute time from 91 unique devices, most of which were running episodically. Figure 6b shows that although the median GPU time contributed by volunteers across all devices was $\approx 1.5$ days, some participants ran the training script on several devices, attaining more than 200 hours over the duration of the experiment. With the exception of the start and the end of the collaborative run, the number of simultaneously active devices mostly varied between 15 and 35 depending on the local time. There was less activity in the last 3 days, likely because the volunteers could see that the model has converged on a public Weights & Biases [101] dashboard.

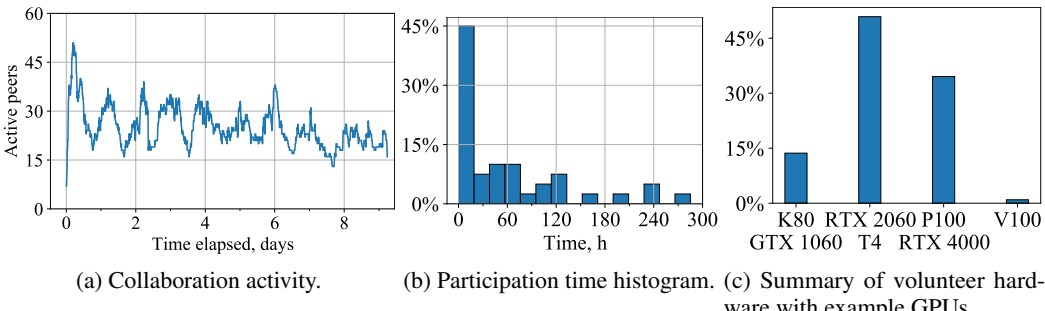

(a) Collaboration activity.

(b) Participation time histogram.

(c) Summary of volunteer hardware with example GPUs.

Figure 6: Collaborative experiment summary.

As depicted in Figure 6c, individual device performance varied significantly among the collaborators. Along with the resources provided by participants, we also used 16 preemptible single-GPU cloud T4 instances for training. We have estimated that the average volunteer device consumed 6.95 GB of network traffic per hour of training. While this bandwidth usage is by no means insignificant, it is comparable with cloud gaming [102] or high-quality video streaming [103].

The model converged after 8 days of training, which is 1.8x as fast as regular distributed training with 8 V100 GPUs that we ran as a baseline; Figure 7 displays the convergence plots for both setups. At the same time, the stepwise learning curves of the two runs were virtually identical (see Appendix I.6), which supports our hypothesis that training with DeDLOC is equivalent to a regular large-batch SGD.

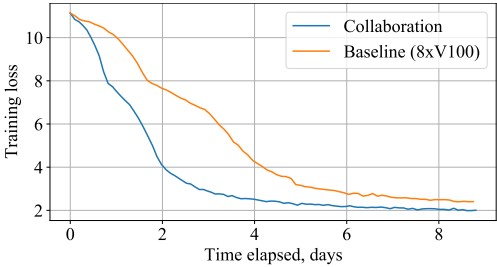

| Model | Wikiann F1 | NCC Accuracy |
|---|---|---|
| bnRoBERTa | $82.32 \pm 0.67$ | $80.94 \pm 0.45$ |
| IndicBERT | $92.52 \pm 0.45$ | $74.46 \pm 1.91$ |
| XLM-R | $96.48 \pm 0.22$ | $90.05 \pm 0.38$ |
| sahajBERT | $95.45 \pm 0.53$ | $91.97 \pm 0.47$ |
| sahajBERT-XL | $\mathbf{96.59 \pm 0.26}$ | $\mathbf{92.91 \pm 0.43}$ |

Figure 7: Training progress of sahajBERT.          Table 2: Downstream evaluation results.

Finally, we compared the Bengali language representations of sahajBERT with those of other pretrained models on several downstream applications. The first model is XLM-R Large [9] — a cross-lingual Transformer-based masked language model that was pretrained on 100 languages and remains a strong baseline for multilingual representation learning. Similarly to sahajBERT, the second model, IndicBERT [104], is also based on the ALBERT architecture; however, it was pretrained on 12 languages, including Bengali and Indian English. The third model, bnRoBERTa [105], is a RoBERTa architecture trained on a monolingual Bengali corpus. We evaluate the model quality on two tasks: WikiANN [106] named entity recognition dataset and Soham News Category Classification benchmark from IndicGLUE [104]. For a detailed description of the setup, refer to Appendix I.8.

As shown in Table 2, sahajBERT performs comparably to three strong baselines despite being pretrained in a heterogeneous and highly unstable setting. Notably, our collaboratively trained model outperforms two specialized monolingual baselines and demonstrates competitive results to XLM-R Large, even though the latter has significantly more parameters (560 million instead of 17 million) and was trained on five hundred high-performance data center GPUs instead of tens of low-cost or even free-tier accelerators. This result confirms previous findings on the benefits of parameter sharing that were made by authors of ALBERT. Also, it highlights one additional advantage of such architectures: specifically, one can train a high-quality representation model in a communication-constrained setting (for instance, over the Internet) without facing noticeable data transfer bottlenecks.

# 5 Conclusion

In this work, we proposed DeDLOC — a collaborative deep learning approach that enables large-scale collective distributed training on whichever computers available to participants, regardless of hardware and network limitations. We demonstrated with several experiments that this is a viable approach that maintains its efficiency in a broad range of conditions. Finally, we report the first real collaborative training run of such a scale and share our findings on volunteer activity to pave the road for similar experiments in the future.

An essential property of collaborative training is its environmental impact. While all distributed training experiments have a negative impact due to carbon emissions [107], DeDLOC has one unique advantage. Due to the ability to utilize heterogeneous low-end devices, it can prolong the effective lifespan of existing computers. We discuss other aspects of environmental impact in Appendix J.

One issue that needs to be addressed before starting collaborative experiments is the need to gather a community of volunteers. Although our proposed authentication mechanism (see Appendix I.5) allows acknowledging participants for their contributions (briefly discussed in Appendix I.2), the best approach to recruit volunteers is an open question: one needs to take into account both the resources of community members and their motivation for training a specific model.

## Acknowledgements

We thank Stas Bekman, Dmitry Abulkhanov, Roman Zhytar, Alexander Ploshkin, Vsevolod Plokhot-nyuk and Roman Kail for their invaluable help with building the training infrastructure. Also, we thank Abhishek Thakur for helping with downstream evaluation and Tanmoy Sarkar with Omar Sanseviero, who helped us organize the collaborative experiment and gave regular status updates to the participants over the course of the training run. Finally, we thank the anonymous reviewers for their feedback on the content and the presentation of our paper.

In addition, authors would like to thank the students of Yandex School of Data Analysis who volunteered to participate in preliminary experiments.

We kindly thank all participants of the Neuropark community[4] who contributed to sahajBERT training. Below, we list the community members who agreed to provide their name for this paper: Aakash Gupta, Aninda Goswamy, Anjali Prasad, Anurag Singh, Arijit Sarkar, Chirranjit Ghosh, Debajit Mallick, Ibraheem Muhammad Moosa, Ishan Bagchi, Khalid Saifullah, Laxya Agarwal, Manan Dey, Mir Ali, Mrinal Mathur, Nilavya Das, Preetha Suri, Priyadarshan Sarkar, Sagnik Roy, Sahil Saha, Sanjeev Kumar, Sanskar Upadhyay, Shyam Sunder Kumar, Soumi Kaibartya, Subhranil Sarkar, Sujit Pal, Syed Modassir Ali, Tanmoy Sarkar, and Vaishali Pal.

Training sahajBERT-XL and hybrid GPU-TPU experiments were made possible by John Kintree, Debajit Mallick, Avijit Saha, Ishan Bagchi, Nilavya Das, Priyadarshan Sarkar, Sagnik Roy, Eduard Pokonechnyy, Arina Ruck. Finally, we would like to acknowledge Tanmoy Sarkar for setting up the backbone peer for sahajBERT-XL on his server and contributing to the evaluation codebase.

The computational resources for internal experiments on cloud instances were provided by the Amazon Research Awards program.

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
