## Contributions

### Conceptual

**Michael Diskin** derived the optimization problem for adaptive averaging.

**Max Ryabinin** designed and led the research.

**Thomas Wolf** initially proposed to run collaborative training with the community participants.

**Gennady Pekhimenko** supervised the work from the systems design point of view.

### Technical

**Alexey Bukhtiyarov** implemented the core large-batch decentralized optimization procedure.

**Dmitry Popov** implemented the support of client mode and auxiliary CPU peers for training.

**Michael Diskin** implemented and conducted the ALBERT pretraining experiments.

**Anton Sinitsin and Dmitry Pyrkin** implemented and conducted the SwAV pretraining experiments.

**Quentin Lhoest** designed and implemented the training data streaming logic.

**Alexander Borzunov and Lucile Saulnier** proposed and implemented the authentication protocol.

**Max Ryabinin** provided the initial code for cloud-based ALBERT pretraining.

**Maxim Kashirin, Denis Mazur, and Ilia Kobelev** implemented the libp2p integration.

**Max Ryabinin** supervised the development of the project and reviewed the code of contributions.

### sahajBERT

**Michael Diskin, Alexey Bukhtiyarov, and Dmitry Popov** created the notebooks with instructions.

**Lucile Saulnier** built the tokenizer for sahajBERT and implemented Bengali-specific preprocessing.

**Michael Diskin, Lucile Saulnier, Max Ryabinin, and Alexander Borzunov** managed the running sahajBERT experiment, monitored its performance, answered the questions of participants, and investigated the occurring errors.

**Albert Villanova del Moral** implemented and conducted downstream finetuning experiments.

**Michael Diskin** created the dashboards and implemented continuous reporting of experiment metrics.

**Alexey Bukhtiyarov** added automatic model state fetching and pushing to Model Hub.

**Yacine Jernite** helped to find the Neuropark community that was interested in collaborative training.

### Writing

**Max Ryabinin** composed the initial structure of the paper, wrote its abstract and the introduction.

**Max Ryabinin, Dmitry Popov, and Alexey Bukhtiyarov** discussed the distributed training, volunteer computing, and federated learning aspects of related work, respectively.

**Max Ryabinin, Lucile Saulnier, and Yacine Jernite** wrote the conclusion of the work.

**Michael Diskin** discussed the use of group-based All-Reduce for training in larger collaborations.

**Michael Diskin** conducted the cost analysis of different distributed training approaches.

**Maxim Kashirin, Denis Mazur, and Ilia Kobelev** described methods for NAT traversal along with peer-to-peer networking.

**Max Ryabinin, Michael Diskin, and Anton Sinitsin** outlined decentralized data streaming.

**Yacine Jernite** assessed the environmental implications of DeDLOC.

**Gennady Pekhimenko and Thomas Wolf** helped improve the general presentation of the work.

**Max Ryabinin, Michael Diskin, and Gennady Pekhimenko** edited the final version of the paper.

## Supplementary Material

## A  Federated learning

Federated learning (FL) is an approach that trains the model on decentralized data stored on many devices without sharing private training data [57]. This scenario is currently gaining more popularity with the rising awareness of data privacy and emerging legal constraints, such as GDPR. Similarly to our setting, FL systems must deal with unreliable heterogeneous hardware. However, their main goal is to ensure the data privacy, which often leads to sacrifices in terms of efficiency.

Most practical FL systems utilize a central parameter server that aggregates local gradients from workers and updates the global model. As we increase the number of workers, the total system performance becomes bounded by the throughput of this server. The problem is exacerbated by secure aggregation protocols [108, 109] that further increase the communication overhead to ensure data privacy. To account for these limitations, production FL systems perform each update using only a small random subset of peers, while the rest remain idle [58]. Contrary to this, our goal is to maximize the training performance by running computations on all peers.

Another recent line of work explores federated learning algorithms with a decentralized communication topology. Maintaining data privacy in these conditions also requires specialized techniques that introduce communication overhead. For instance, [59] proposes a system where workers cannot share parameters directly, relying on a secure peer-to-peer knowledge distillation instead.

The above discussion makes it clear that the purpose of the federated learning is orthogonal to ours: we aim to train the global model on publicly available data and achieve the best possible performance.

## B  Optimal averaging strategy via linear programming

Recall that DeDLOC finds the optimal communication strategy by solving the following problem:

$$
\begin{aligned}
\max_{a,g,c} \quad & \min\left( \frac{\sum_{i=1}^{n} s_i \cdot c_i}{B},\ \frac{\min_i \sum_j g_{ji}}{P} \right) \\
\text{s.t.} \quad & g_{ij} \leq \min_{k:c_k=1} a_{ki} & \forall i,j \\
& \sum_{j \neq i} (a_{ji} + g_{ji}) \leq d_i & \forall i \\
& \sum_{j \neq i} (a_{ij} + g_{ij}) \leq u_i & \forall i \\
& a_{ij} + g_{ij} \leq t_{ij} & \forall i,j \\
& a_{ij} \geq 0\ \&\ g_{ij} \geq 0\ \&\ c_i \in \{0,1\} & \forall i,j
\end{aligned}
\tag{2}
$$

Here, $a_{ij}$ denotes the fraction of network throughput allocated to sending gradients from peer $i$ to peer $j$ for aggregation, $g_{ji}$ is the corresponding fraction for returning the averaged tensors back to sender, and $c_i$ is a binary indicator that represents whether or not peer $i$ computes gradients. The remaining variables are parameters that denote peer compute performance $s_i$, maximum download and upload speeds ($d_i$ and $u_i$ respectively) and regional limitations of peer-to-peer throughput ($t_ij$). Finally, $B$ denotes the global target batch size per step and $P$ is the number of model parameters.

As stated earlier in Section 3.2, the DeDLOC peers need to find the optimal strategy during each averaging round. As such, we must ensure that the procedure for solving (2) does not introduce any significant overhead. To that end, we reformulate the problem as a linear program by means of several consecutive reductions, which are described below.

**Max-min LP reduction.**  First, we replace the original max-min objective with a linear one by following the technique described in [110]: we maximize a new surrogate variable $\xi$ and replace the inner $\min$ by two additional constraints:

$$
\begin{aligned}
\max_{a,g,c} \quad & \xi \\
\text{s.t.} \quad & \xi \leq \frac{\sum_{i=1}^{n} s_i \cdot c_i}{B} \\
& \xi \leq \frac{\sum_j g_{ji}}{P} & \forall i
\end{aligned}
\tag{3}
$$

**Binary to LP relaxation.** Second, we must account for the binary variable $c_i$. From a formal perspective, using these indicators transforms our problem into a binary mixed-integer program with a combinatorial worst-case complexity. However, for this specific problem, it is possible to rewrite the constraints in such a way that $c_i$ can be treated as a continuous variable $0 \le c_i \le 1$:

$$\forall i, j, k \in 1 \dots n \quad g_{ij} \le a_{ki} + (1 - c_k) \cdot d_i \tag{4}$$

For $c_k = 1$, the above equation (4) is exactly equivalent to the original constraint $g_{ij} \le \min_{k:c_k=1} a_{ki}$. In turn, setting $c_k < 1$ for some $k$ effectively removes the corresponding peer $k$ from the $\min$ operator, allowing participant $i$ to aggregate tensors with up to its maximum download speed $d_i$ instead of waiting for peer $k$. The $d_i$ factor in (4) can be replaced with any large positive number as long as the constraint (4) is not saturated for $c_k{=}0$. In practice, $c_k \ne 1$ corresponds to peer $k$ **not** computing gradients, but still assisting in gradient aggregation.

Applying the two above reductions, we get the following linear program:

$$
\begin{array}{lll}
\max\limits_{a,g,c} & \xi & \\
\text{s.t.} & \xi \le \sum_{i=1}^{n} s_i \cdot c_i \ / \ B & \\
& \xi \le \sum_j g_{ji} \ / \ P & \forall i \\
& g_{ij} \le a_{ki} + (1 - c_k) \cdot d_i & \forall i, j, k \\
& \sum_{j \ne i} (a_{ji} + g_{ji}) \le d_i & \forall i \\
& \sum_{j \ne i} (a_{ij} + g_{ij}) \le u_i & \forall i \\
& a_{ij} + g_{ij} \le t_{ij} & \forall i, j \\
& a_{ij} \ge 0 & \forall i, j \\
& g_{ij} \ge 0 & \forall i, j \\
& 0 \le c_i \le 1 & \forall i
\end{array}
\tag{5}
$$

To avoid additional synchronization steps, each peer within DeDLOC solves the above problem (5) independently using the interior point solver [111]. Based on the obtained solution, peer $i$ will aggregate a fraction of gradients proportional to its effective throughput:

$$\text{fraction}_i \propto \frac{\min_j g_{ij}}{\sum_k \min_j g_{kj}}. \tag{6}$$

Furthermore, if $c_i \ne 1$, the corresponding participant will disregard its local gradients. In the future, it may be possible to allow such peers to contribute partial gradients akin to [39]. However, we leave this investigation to future work.

For certain collaboration compositions, there can be multiple optimal strategies with equal training throughputs. To ensure that all participants act according to the same strategy, we require each peer to solve (5) using a deterministic interior point algorithm with globally consistent hyperparameters [112].

Another practical consideration is that some peers are unable to compute gradients or perform aggregation (for instance, due to networking issues described in Section 3.3). To account for these limitations, we exclude such peers from aggregation in $\frac{\sum_{i=1}^{n} s_i \cdot c_i}{B}$ and $\frac{\sum_j g_{ji}}{P}$ terms for compute and network resources, respectively.

## C   Fault tolerance

In practice, using DeDLOC with large collaborations will eventually require dealing with node failures. If the failures are rare, it is possible to restart the failed steps until they succeed. However, if the collaboration size increases, this strategy will eventually become impractical.

One possible solution is to replace the global (collaboration-wide) All-Reduce with several parallel operations, which is known as Group All-Reduce [29] or Moshpit All-Reduce [56]. Each operation involves a small independent group of $m$ peers, whereas the groups themselves are formed in such a way that the collaboration can obtain the global average in a logarithmic number of rounds.

Under this strategy, any failed device will only affect its local group instead of the entire collaboration. Furthermore, each individual group will have a higher success rate, since it contains $m \ll n$ peers.

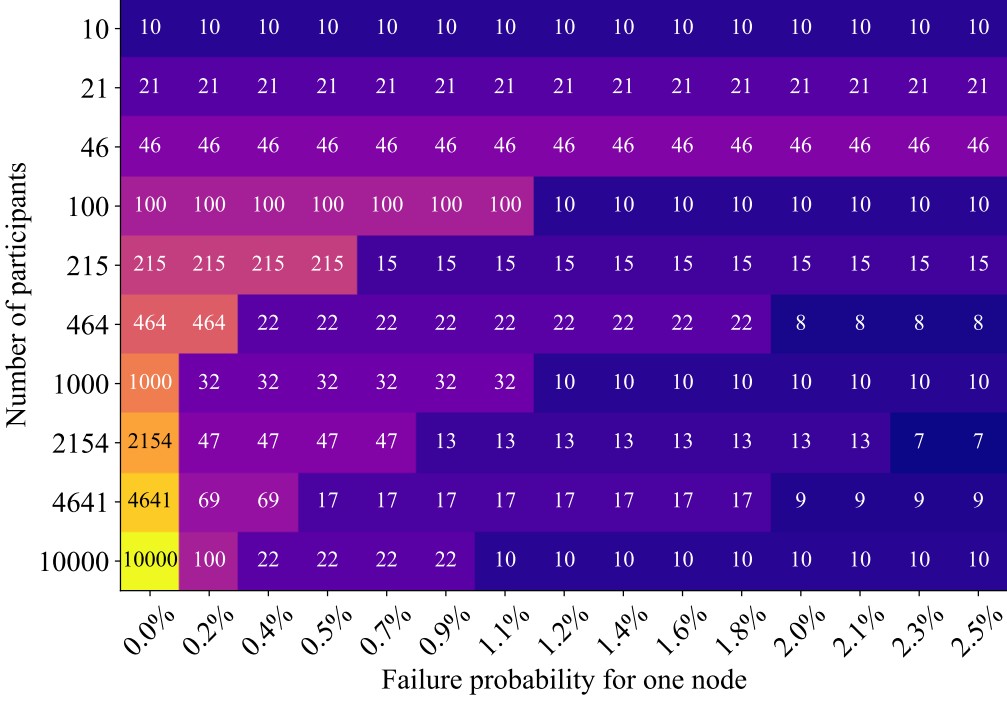

Figure 8: Optimal group size for different collaboration sizes and failure rates.

In turn, the drawback of using group-based All-Reduce is that the collaboration will need $\lceil \log_m n \rceil$ steps to obtain the global average.

We can select the optimal group size by minimizing the *expected* number of iterations required to compute the global average, including both restarts from node failures and the overhead from using Group All-Reduce. For reference, we include the optimal group sizes for typical collaborations and failure rates in Figure 8. In all our experiments, the optimal group size was $m=n$ due to a small number of participants and very rare significant network failures.

# D    Network address translation

Collaborative training, similarly to any other application incorporating peer-to-peer communication, is susceptible to a number of networking issues, among which the most common is the inability to accept incoming connections due to Network Address Translation, or NAT [89]. The primary function of NAT is to separate the address space of the local network from the global address space by dynamically translating addresses and port numbers of outgoing sessions into public endpoints. Therefore, NAT helps deter the rapid depletion of IPv4 addresses and provides additional security by hiding the local network structure from external parties. However, this also means that NAT devices only authorize outgoing connections, since the dynamic mapping of local endpoints makes it impossible to forward incoming packets to the proper internal host.

For the purposes of the current work, NAT devices can be categorized into two groups — cone and symmetric. A cone NAT translates an internal IP address and port to the same globally routable endpoint regardless of the destination host, whereas a symmetric NAT allocates different address mapping for each destination host. In case of UDP traffic, the cone NAT can be traversed using the mechanism of UDP Hole Punching. Briefly put, this technique consists of two stages. During the first phase, peers A and B connect to the same globally accessible rendezvous server using the STUN protocol [113] and exchange their public and private endpoints. The rendezvous server is often called the STUN server by the name of the protocol. At the next step, both peers start sending UDP data packets to each other's endpoints. If A's packet reaches NAT B before B's packet "punches a hole", then it is dropped by the NAT B, but when the B's packet reaches NAT A shortly after this, the outgoing session has already been initiated by A, so the B's request is successfully forwarded to A. If both peers happen to "punch a hole" in their NATs before the arrival of the counterpart's packet, then the connection is established immediately.

For the TCP traffic, hole punching is also possible, though it has to overcome additional API issues that arise because of the client-server paradigm around which TCP was designed. However, peer-to-peer communication over TCP connections is more robust than over UDP, since NAT usually timeouts the UDP port mapping, thus periodical keep-alive messages must be transmitted. As reported in [90], currently almost two thirds of all NAT vendors provide devices which are compatible with TCP hole punching, that is, consistently map private endpoints and do not send back Reset packets to unsolicited requests.

As for the symmetric NAT, only relaying through a third-party proxy can help establish the connection between peers. This is supported with the TURN protocol [91]. If two peers fail to connect via hole punching, they appeal to the TURN server for an interaction through it.

## E  Peer-to-peer network infrastructure

To enable peer-to-peer interactions that can bypass NAT, we can use the libp2p framework [114]. Each peer has a set of multiaddresses that allow other participants to establish a connection. Multiaddress comprises an IP address, an L4 protocol (TCP/UDP) with a port, an optional high-level protocol (QUIC), and a peer identifier. A peer can listen to several transport protocols, but it may have only one identifier.

After peers connect to the network, they can interact with each other via their respective identifiers. There are no dedicated STUN and TURN servers in the libp2p network: their role is played by public participants. The network must contain at least 4 publicly accessible peers to be able to recognize public addresses of newly connected peers. Optimally, these are well-known peers with multiaddresses known to all participants. Upon joining, a new node synchronizes with the DHT used for routing and receives information about other available peers. After that, a peer can interact with other participants using their peer id. If the network can get the public address of the peer, then other participants will be able to connect to it.

If a public address of the peer is not available or two peers are using different transport, the communication can be started by relaying requests via an intermediate participant. Libp2p supports the autorelay feature that allows finding the best relay automatically. When autorelay is enabled, a public peer can serve as a relay for other participants, and a private peer will find the best relay.

## F  Cost analysis

In this section, we provide a detailed cost analysis of several hardware and networking setups that can be used for both tasks described in Section 4, namely, SwAV and ALBERT pretraining.

For simplicity, we only consider temporary resource ownership, i.e., renting GPU-enabled servers instead of building it on-premise. The latter option can be more cost-efficient in the long term, but might be impractical if only a few training runs are required. For the same reason, we do not consider discounts available for committed usage of the same resource over multiple years. As for the rented resources, there are several general hardware categories that we consider:

1. High-performance cloud GPU — dedicated instances with multiple high-end compute accelerators and extremely fast device interconnect.
2. Low-end cloud GPU — single-GPU instances with NVIDIA M60, T4 or P40, linked with a fast (preferably intra-datacenter) network of 10–50 Gb/s.
3. Commodity GPUs — regular desktop-like machines with consumer-grade GPUs, like NVIDIA RTX 2070, 2080 Ti, 3070. On average, they can have higher performance than low-end cloud devices, but lower network throughput (50–200 Mb/s).
4. Volunteer hardware — almost the same class of devices as in the previous section, with the same advantages and disadvantages, but "free" for the experiment organizers.

For a fair comparison, we consider three types of GPU instances: cloud V100, cloud T4 and commodity GPUs from peer-to-peer marketplaces, such as `vast.ai` or `golem.ai`. While several cloud providers offer newer generation GPUs (NVIDIA Ampere), this GPU lineup is still in an active rollout phase, which causes significant price fluctuations. Thus, we base our conclusions on more established generations of GPUs.

In addition to GPU instances, DeDLOC can also benefit from non-GPU servers that act as auxiliary parameter aggregators. The only real requirement for such servers is high network bandwidth. As such, we consider additional resource types:

1. Premium cloud VMs — low-end instances from premium cloud providers. We consider instances with 2 cores, 16GB RAM and 25 Gb/s maximum bandwidth (symmetric).

2. Economy cloud VMs — similar cloud instances (or dedicated servers) from economy cloud providers. For this run, we consider instances with the same 2 cores / 16GB RAM, but only 300–1000 Mb/s symmetric bandwidth (depending on the provider).

3. Volunteer non-GPU devices — in theory, it is possible to run collaborative training entirely on volunteer devices with zero hardware expenses for the organizer. However, we omit this option as it trivializes our cost analysis.

On top of that, all cloud and marketplace instances can be rented in a guaranteed ("on-demand") or a non-guaranteed option. In the latter scenario, the resources are offered at a significant discount, but the resource provider can terminate such instances at any time.

Based on the available resource types and ownership models, we assemble six server fleets with approximately equal training performance in our two experimental setups. For convenience, we order these setups by how difficult they are to operate (easiest-first):

- Single high-end node — 8 x NVIDIA Tesla V100: easiest to operate, but the most expensive option.
- Preemptible high-end node has the same hardware but costs less due to irregular availability, which creates a need for regularly saved checkpoints.
- Distributed nodes — 16 x NVIDIA Tesla T4: homogeneous, require distributed optimization.
- Distributed + preemptible — same but preemptible, can be used with a framework that supports elastic training, such as TorchElastic[52] or Elastic Horovod[53].
- Distributed + heterogeneous — 5x NVIDIA GTX 1080 Ti, 3x RTX 2070, 1x 2070S, 2x 2080, 4x 2080 Ti, 1x 3070. This configuration has lower bandwidth, thus additional CPU-only peers are needed for efficient averaging.
- Collaborative training — for this setup, we assume that the GPUs from the previous setup are available from volunteers. In that case, the only sources of expenses for the organizer are networking and CPU-only nodes.

As one can see in Table 3, using a single high-end node is the most expensive alternative. Switching to multiple lower-end nodes and using non-guaranteed instances reduces the cost by a factor of $\approx$ 3x each. Finally, the volunteer infrastructure is two orders of magnitude cheaper than the high-performance setup. However, some of this price difference is effectively shifted to volunteers. Based on average electricity and networking costs of household Internet connections, we estimate the expense at $9–30 *per volunteer per month*, assuming 16 volunteers with equivalent GPUs. However, actual costs can vary based on the region, time duration and the exact hardware used by each volunteer.

Finally, we want to reiterate that the above setups require different amounts of effort (and expertise). Training on a single high-end node can be done with virtually no code changes in major deep learning frameworks, such as TensorFlow [115] or PyTorch [98]. In contrast, multi-node (and especially elastic) setups require specialized distributed training frameworks and careful performance tuning. Finally, working with volunteer or marketplace instances introduces a new layer of complexity, which is addressed in this paper.

Table 3: Costs of training setups.

| Setup | Instance types | Monthly cost |
|---|---|---|
| Cloud on-demand | 8xV100 | $16,898 |
| | 16xT4 | $5,299 |
| Cloud preemptible | 8xV100 | $5,133 |
| | 16xT4 | $2,074 |
| Marketplace | 4xCPU+16xGPU | $5,148 |
| Volunteer | 4xCPU | $257 |

**Networking costs.** When done naïvely, training with geographically distributed participants can incur significant networking expenses. For instance, when using preemptible cloud GPUs from a major provider, allocating these GPUs in different regions can incur additional costs of more than $3000 per month, compared to a total hardware cost of $2074 for the same period.

More importantly, using premium non-GPU instances for collaborative training will also incur additional networking costs. Based on our preliminary experiments, a collaborative training setup equivalent to Table 3 would lead to an average networking bill of $5000-6000 per month. Fortunately, it is possible to circumvent this expense by using cloud providers that do not charge additional costs for network traffic. These providers typically offer less reliable instances with lower maximum bandwidth, which is not a significant issue for DeDLOC.

As a general recipe for reproducing our experiments, we recommend using one of the two setups. When running experiments internally, one can use any major cloud provider as long as all instances are *configured to avoid cross-regional networking costs* (e.g. use internal address space). In contrast, when training with actual volunteer devices, we recommend using cloud providers without additional networking charges or existing server infrastructure.

## G   Convergence analysis

As discussed in Section 3.1, DeDLOC updates parameters only after accumulating the gradients for the target number of samples from up-to-date peers. However, due to network-related delays, peers can process more samples than required in some cases. Thus, we can analyze DeDLOC as a regular SGD with varying batch sizes, which allows us to adapt the existing convergence bounds from the optimization literature. More formally, consider a standard optimization problem

$$\min_{x \in \mathbb{R}^n} f(x), \tag{7}$$

which is solved by SGD. We denote the gradients for step $k$ as $\mathbb{E}[g^k|x^k] = \nabla f(x^k)$ and the corresponding update as $x^{k+1} = x^k - \gamma_k g^k$.

Denote the variance of a single stochastic gradient as $\mathbb{E}\left[\left(\nabla f(x^k, \xi_i^k) - \nabla f(x^k)\right)^2 | x^k\right] \leq \sigma_0^2$ and the target batch size as $m$. At step $k$, DeDLOC will accumulate gradients from $m_k \geq m$ samples:

$$g^k = \frac{1}{m_k} \sum_{i=1}^{m_k} \nabla f(x^k, \xi_i^k). \tag{8}$$

Thus, the gradient averaged over a minibatch of $m_k$ i.i.d. samples will have the following variance:

$$\mathbb{E}\left[\left(g^k - \nabla f(x^k)\right)^2 | x^k\right] = \frac{1}{m_k^2} \sum_{i=1}^{m_k} \mathbb{E}\left[\left(\nabla f(x^k, \xi_i^k) - \nabla f(x^k)\right)^2 | x^k\right] \leq \frac{1}{m_k^2} \sum_{i=1}^{m_k} \sigma_0^2. \tag{9}$$

Because $m_k \geq m$,

$$\frac{1}{m_k^2} \sum_{i=1}^{m_k} \sigma_0^2 = \frac{\sigma_0^2}{m_k} \leq \frac{\sigma_0^2}{m}, \tag{10}$$

which allows us to reuse the existing SGD convergence bounds from the optimization literature [116, 117]. For instance, we can use Theorem 5 from [116] and plug in $\frac{\sigma_0^2}{m}$ as gradient variance (with notation also from [116]), getting the following result:

$$\mathbb{E}f(\bar{x}_T) - f^\star + \mu\mathbb{E}|x_{T+1} - x^\star|^2 \leq \min\left\{64LR^2 \exp\left[-\frac{\mu T}{4L}\right] + \frac{36\sigma_0^2}{\mu m T}, \frac{2LR^2}{T} + \frac{2\sigma_0 R}{\sqrt{mT}}\right\}. \tag{11}$$

# H  Decentralized data streaming

In this section, we propose a generalization of our data streaming approach described in Section 3.3 to a setting without any central data storage. Namely, we offer a way to to distribute large datasets across all participants by sharding the examples in the same manner that was used previously.

Specifically, this approach is based on the notion of a local buffer combined with the decentralized metadata storage enabled by the DHT. When a peer joins the experiment, the training process allocates a buffer for several chunks on a local high-capacity storage device (HDD/SSD) available to that peer; the number of chunks is determined by the participant and depends on the hardware capabilities of their computer. Then, in order to procure training data, the peer queries the DHT to find the shards that are stored on the least number of other peers. Assuming that the number of shards does not exceed several thousand, this search can be done by a simple linear-time lookup of all keys without any significant performance drawbacks. After finding such shards, the training process randomly chooses one shard from this set and downloads it from another peer. When the download is complete, the participating node trains on batches from this shard and stores it for later use by other members of the network. The training process repeats such iterations; if the local buffer becomes full at any point, the shards with the highest replication factor are evicted in favor of new data.

The decentralized approach to data streaming has two immediate benefits. First, similarly to distributed training, this approach reduces the load on a single server (or the content delivery network), which might result in significant savings for large-scale experiments that use datasets hosted by cloud providers. Second, even when the data is hosted by organizers of the collaborative experiment, its size might be too large to prevent efficient storage and sharing without investments in specialized infrastructure, which is often quite expensive as well. Storing small portions of the dataset on the computers of participants allows circumventing both issues by distributing the load among all peers. However, we note that the above approach was not implemented for our current experiments; this section is intended to serve as a description of future work.

# I  Collaborative experiment setup

## I.1  Instructions for participants

All communication with volunteer contributors took place on a group instant messaging platform. Prior to launching the experiment itself, we used this platform to communicate with Bengali speakers in order to validate the language-specific elements of the model, such as the normalization component of the tokenizer and the sentence splitter tool.

Then, for the collaborative training, we first sent several introductory messages before the event to explain what the event will consist of. Then, we sent a message the day before and a message on the event's launch day with instructions on how to join the training run. Lastly, we sent daily messages to report the current status of the event. The content of the first such message can be found in Figure 9.

In this message, the volunteers were invited to:

1. Submit their Hugging Face usernames;
2. Once added to the allowlist, join the training via notebooks provided by the organizers. After checking that the connection was established and that the GPU was available, participants had to run the notebook and fill in their credentials for the Hugging Face authorization API.

## I.2  Measurement of volunteer contributions

To let participants follow their own contributions as well as the overall training effort, they were given access to real-time Weights&Biases dashboards. Each participant could see their personal contributions with the total number of training examples they processed, as well as how much time they contributed and the loss function dynamics of their local models. The volunteers also could compare their contributions: in particular, participants with more computational resources could see the impact they had by comparing the number of samples per second they contributed with other runs. Finally, at the end of the event, a leaderboard of the ones with the highest number of contributed examples was shared with everybody to congratulate the participants.

Hi @everyone! We're starting the Collaborative Training Experiment now! Here is some important information:

**How to participate?**
1. As a reminder, you need to provide your Hugging Face username to be able to participate. For the current participants, @Tanmoy already gathered this list (thank you @Tanmoy!). For new participants, please join *#albert-allowlist* and add your username. Someone from the team will add you to the allowlist. If you see a ⏳ reaction, we're on it! If you see a ✅, you should be added by then. Feel free to reach out to @Omar Sanseviero, @Mike Diskin, @Quentin Lhoest, @Lucile Saulnier or me if you don't have access.
2. You can join the training with:

- **Colab**: link

- **Kaggle**: link
  This option provides you a P100 and lasts longer than Colab. This requires a Kaggle account. You must **enable Internet access and switch kernel to GPU mode** explicitly. If it is stuck at "installing dependencies" for over 5 minutes, it means you changed the session type too late. Simply restart with GPU/Internet enabled and it should work just fine.

Please do not run multiple GPU instances on the same service! You can use Kaggle in one tab and Colab in another, but avoid having two Colab GPU instances at the same time.

Local run: if you have a local GPU and you're tech-savvy. We will keep you informed when this option is available. Stay tuned!

Feel free to ask any questions in *#albert-bengali-training* channel and reach out to us (at the right you can see the members of the Yandex and HF teams).
In the following dashboard you can track the status of training: link

Thank you all for participating and let us know if you have any questions!

Figure 9: The message sent to participants at the event launch. Parts in grey refer to external links.

Although this scheme proved to be highly engaging, it could be improved by also acknowledging the peers that do not contribute the GPU resources but are still very helpful to the collaboration. For example, CPU-only peers with faster network connections can be rewarded for successful averaging rounds and compared between each other in terms of the total number of averaged parameters. Also, to encourage long-term involvement and to increase the stability of the experiment, it might be possible to maintain a list of volunteers with the longest participation time without interruptions.

### I.3   Tokenizer

For this experiment, we used the architecture of the ALBERT model [7]; the authors of the original work have chosen the unigram language model [118] token segmentation algorithm that allows transforming a raw text into subword units based on a fixed size vocabulary of 30k tokens. In order to use the tokenizer that is adapted to the Bengali language, we created a new tokenizer using the Tokenizers library [119].

This tokenizer is composed of:

- Several normalizations adapted to the Bengali language: NMT normalization, NFKC normalization, removal of multiple spaces, homogenization of some recurring unicode characters in the Bengali language and lowercasing;

- Specific pre-tokenization rules to condense the vocabulary: we split on whitespaces and replace them with an underscore character "▁" (U+2581), we also isolate all punctuation and digits from any other characters;

- A Unigram language model as a segmentation algorithm with a 32k tokens vocabulary, trained on the deduplicated Bengali subset of OSCAR [100];

- A template postprocessor, allowing a special token "[CLS]" to be included at the start of an example, as well as a special token "[SEP]" to separate two segments and to denote the end of sequence.

### I.4   Dataset streaming

Streaming the data to each participant allows them to start training immediately, since the participants do not have to download the full dataset before launching the training. More specifically, the examples from the dataset can be downloaded progressively as training goes. To do so, we used the datasets library [120]. It enabled streaming of Wikipedia and OSCAR, as well as shuffling, on-the-fly processing and mixing of the datasets.

For the experiment, we use the Wikipedia and OSCAR Bengali datasets. Both datasets are split in shards, respectively in the Parquet and GZIP-compressed raw text formats. Information about the datasets is given in Table 4. The participants download the examples from those files during training, since it is possible to iterate row group by row group from Parquet files and line by line from compressed text files.

The Bengali Wikipedia dataset is based on the 03/20/2021 Wikipedia dump. The data was processed using the Wikipedia processing script of the datasets library in early April of 2021. Each example contains the content of one full article, cleaned from markup and sections such as references.

Table 4: Sizes of the Bengali Wikipedia and OSCAR datasets used for training.

|  | Wikipedia | OSCAR |
|---|---|---|
| Uncompressed size | 657MB | 6.2 GB |
| Documents | 167,786 | 1,114,481 |
| Shards | 10 | 4 |

To shuffle the datasets, we make each participant iterate over the shards in random order. Then, a shuffle buffer of size $S = 10000$ is used, which is compatible with the progressive download of examples. We use a shuffle buffer, because we do not want the participants to download entire shards in the beginning of training just for shuffling.

Sentence splitting, tokenization and preprocessing for next sentence prediction are applied to the examples in an online manner. Since these steps are several orders of magnitude faster than forward and backward passes of the model, they have no significant impact on the training performance.

### I.5   Participant authentication

Since our experiment was an open collaboration, we chose to set up an authentication system allowing only the people motivated by the final result of the model to join the training. Allowlisting seemed to be the most suitable solution to this need. We therefore distinguish between three types of actors in the distributed network:

- **Central server's moderators**: people who start the experiment, maintain the allowlist and know how to join the training. They have a key pair $(public\_key_{auth}, private\_key_{auth})$ hosted on the central authentication server. In this protocol, the role of the central server is threefold: 1) to verify the identity of a collaborator by requesting the confirmation of an identity provider website, 2) to verify that this collaborator is allowlisted and 3) to distribute access passes to authorized collaborators. Peers have a secure HTTPS-based communication channel with this server in order to protect the data;

- **Digital identity provider**: an entity which is able to create digital identities via a website. In order to create the allowlist, moderators have asked the collaborators to have a digital identity on the identity provider website. This is useful to prevent bots and potential attackers from joining the training and give the moderators opportunity to acknowledge the contribution of each collaborator. In our setup, each identity linked to a username can be claimed by a login and a password owned by one collaborator;

- **Collaborators / Peers**: people who wish to make their computing resources available for collaborative training. Each peer $i$ in the network has a key pair $(public\_key_i, private\_key_i)$. They also have a digital identity on an identity provider website.

The following procedures aim to prevent 1) that a non-allowlisted collaborator can interact with the members of the collaborative training and 2) that a malicious actor could claim to be an allowlisted collaborator:

**Joining the network:** To join the collaborative training, a peer $i$ must request an access pass from the authorization server. To grant the access pass, the authorization server asks the digital identity provider if the peers are who they claim to be. If the entity provider confirms the peer identity, the authorization server checks that the username appears in the allowlist. If these two steps are verified, the authorization server creates an access pass, otherwise it rejects the peer's request. The access pass is temporary and contains the following information:

- The endpoint of a peer already present in the network (a starting point for joining the network);

- An access token $access\_token_i$ composed of a peer's username, its public key $public\_key_i$, and the expiration date of its access pass. The token is signed with the private key $private\_key_{auth}$;

- The public key $public\_key_{auth}$.

With this access pass, the peer can make requests and respond to them in the decentralized network. After expiration, the peer may repeat this procedure to get a new token.

**Making requests:** Alice wants to make a request to Bob. In order for her request to be processed by Bob, we require Alice to include several additional information in her request: 1) her access token $access\_token_{Alice}$, 2) receiver's public key $public\_key_{Bob}$, 3) the current time, 4) a set of random bytes (denoted as *nonce*) that is supposed to be unique for each request and 5) a signature of the request contents and the additional information made with $private\_key_{Alice}$. With this information, Bob considers that a request is not legitimate and should not be processed if one of the following cases occurs:

- Alice's access token $access\_token_{Alice}$ is invalid (its signature does not match $public\_key_{auth}$) or expired;

- The signature of the request does not match $public\_key_{Alice}$ (stored inside $access\_token_{Alice}$);

- The request's current time field differs from the Bob's current time by more than $N$ seconds;

- The nonce has already been used during the last $2N$ seconds;

- The recipient's public key field does not match the real $public\_key_{Bob}$.

These checks protect the exchange against eavesdropped request reuse and man-in-the-middle attacks, because Bob is sure that 1) Alice is specified in the allowlist and her authorization is still valid, 2) the request was created by Alice and could not have been modified by someone else, 3) Bob is the recipient of the request, 4) the request is not repeated by someone who eavesdropped a previous one.

**Responding to requests:** When Bob responds to Alice, we also require Bob to include several additional information in his response: 1) his access token $access\_token_{Bob}$, 2) the nonce sent with Alice's request and 3) a signature of the response contents and the additional information made with $private\_key_{Bob}$. In the same way as above, a response is not considered valid by Alice if:

- Bob's access token $access\_token_{Bob}$ is invalid or expired;

- The signature of the response does not match $public\_key_{Bob}$ (stored into $access\_token_{Bob}$);

- The nonce does not match the nonce stored into Alice's request;

- The sender's public key field does not match the real $public\_key_{Bob}$.

If the response does not check any of the above cases, Alice is sure that 1) Bob is specified in the allowlist and still has valid access, 2) the response was sent by Bob and could not be modified, and 3) it is the response to the request associated with this nonce. Therefore, an eavesdropped response can't be replayed for another request and a man-in-the-middle attacker can't replace the response content.

## I.6 Stepwise learning curves

As one can see on Figure 10, collaborative training is nearly equivalent to regular data-parallel training in terms of the total number of SGD updates. The slight difference between the two curves is likely due to random variation, though it can also be explained by the fact that DeDLOC uses slightly larger batches due to network latency. In other words, some peers will aggregate a few extra gradients between the moment when the collaboration accumulated 4096 samples and the moment when every peer enters the gradient averaging stage.

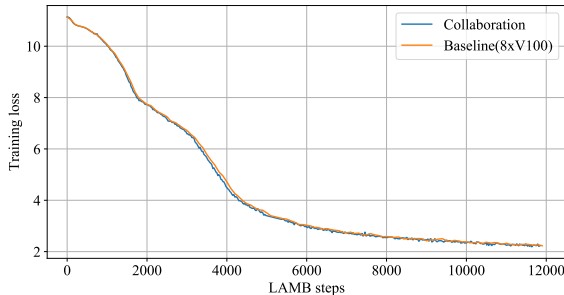

Figure 10: Stepwise training progress of DeDLOC and regular distributed training.

## I.7 Training sahajBERT-XL with hybrid GPU + TPU resources

To better explore the practical ramifications of collaborative training, we asked volunteers to train a larger model on the same task as the original sahajBERT. We refer to this model as sahajBERT-XL, as it has approximately the same size as ALBERT-xlarge [7]: more specifically, the new model has $d_{model} = 2048$, $n_{layers} = 24$ and three additional architecture modifications:

- **Pre-normalization:** the layer normalization [121] was moved to the beginning of each Transformer layer, as in pre-activation residual networks [122]. According to prior work, this modification stabilizes the training process in several Transformer applications [123, 124].

- **GeGLU activations:** the new model replaces GeLU activation functions with their gated counterparts known as GeGLU, which were shown to improve the performance of Transformer models [125, 126]. However, unlike [125], sahajBERT-XL uses the same number of GeGLU units as in ALBERT-xlarge, which results in 17M additional parameters.

- **Rotary embeddings:** instead of learned absolute positional embeddings, we equip sahajBERT-XL with rotary embeddings [127] that were recently demonstrated to improve training stability of large language models [128].

The final model had 72.5M parameters, which is ≈4 times more than for original sahajBERT. To reduce the computational requirements of sahajBERT-XL pretraining, we initialized it with Net2Net conversion [129] from the original sahajBERT checkpoint after 10,000 training steps. Because of architectural differences, we needed to manually remove the learned positional embeddings and create a new set of GeGLU parameters, which were initialized by copying the existing pre-GeLU parameters and adding Gaussian noise with the variance of $10^{-3}$. We increased the training batch size to 16,384 and used the corresponding learning rate schedule from [49]. Before training, we reaccumulated the LAMB optimizer statistics by running 500 steps with a zero learning rate and setting the training schedule to step 3,125, which corresponds to the end of the warmup stage.

Despite using this shortcut, training sahajBERT-XL would still require over 3 months using 8 V100 GPUs. To alleviate this problem, we requested volunteers to use both GPU and preemptible TPU (v2 and v3) instances available in several free-tier cloud providers. As a result, a community of 14 volunteers was able to train sahajBERT-XL in 22 days.

However, training in a hybrid GPU-TPU "cluster" has proven challenging due to different mixed precision capabilities. Specifically, the available GPU instances could train in float32 and float16 formats, while the TPU cores support float32 and **b**float16. Unfortunately, training in float16 on GPU and bfloat16 on TPU caused the model to consistently diverge both with Net2Net initialization and when training from scratch: to combat this issue, we switched TPU computations to float32 while keeping GPU ones in float16. Despite this, a TPUv3-8 peer still outperformed any single GPU node.

Table 5: Hyperparameter values used for model evaluation.

| Task | Model | Learning rate | Input length | Batch size |
|------|-------|---------------|--------------|------------|
| | XLM-R | $10^{-5}$ | 256 | 8 |
| | IndicBERT | $3 \cdot 10^{-5}$ | 256 | 64 |
| NER | bnRoBERTa | $3 \cdot 10^{-5}$ | 512 | 64 |
| | sahajBERT | $10^{-5}$ | 128 | 32 |
| | sahajBERT-XL | $3 \cdot 10^{-5}$ | 256 | 64 |
| | XLM-R | $10^{-5}$ | 128 | 8 |
| | IndicBERT | $3 \cdot 10^{-5}$ | 128 | 128 |
| NCC | bnRoBERTa | $3 \cdot 10^{-5}$ | 128 | 64 |
| | sahajBERT | $3 \cdot 10^{-5}$ | 64 | 64 |
| | sahajBERT-XL | $10^{-5}$ | 128 | 64 |

Using the techniques described above, the volunteers were able to train a model that outperforms both the baselines and the original sahajBERT model on both downstream tasks (see Table 2). However, due to the significant computational requirements of sahajBERT-XL, we were only able to train the model once without proper hyperparameter sweeps and ablation analysis. Thus, we believe that future research will reveal more efficient strategies for training with hybrid hardware accelerators.

### I.8 Evaluation

We compare sahajBERT with three other pretrained language models: XLM-R [9], IndicBert [104], and bnRoBERTa [105]. For downstream evaluation, we use two tasks from the Indic General Language Understanding Evaluation (IndicGLUE) benchmark [104]: named entity recognition (NER) with the balanced train-dev-test splits version [130] of the original WikiANN dataset [106] and news category classification (NCC) with the Soham News Article dataset [104].

Each model was finetuned and evaluated as follows:

1. For each combination of learning rate in (1e-5, 3e-5) and the maximum input length in (64, 128, 192, 256, 512), we finetuned the model on each task and computed the validation set metrics to find the best hyperparameters. We computed the F1 score for NER and accuracy for NCC;

2. For the best configuration, we computed the metrics of the corresponding model on the test set. We repeat this step three times for different random seeds, reporting the mean and the standard deviation of metrics.

All finetuning experiments were run using the Adam [131] optimizer with the weight decay fix [132], weight decay of 0.001, and the linear decay learning rate schedule. Finally, each model was trained for a maximum number of 20 epochs and stopped earlier if the loss on the validation set did not decrease during 3 epochs. The size of the batch was chosen to be as large as possible: we started with a batch size of 128 and then, if necessary, the batch size is decreased until it can be stored in memory. For the exact hyperparameter values, see Table 5.

## J Environmental impact

Recent works have outlined the environmental consequences of training ever larger deep learning models [133, 134] and encouraged authors to report the incurred energy costs [135]. The direction proposed in this work may help in two specific ways. First, while most of the current tools focus on the $CO_2$ cost caused by the training-time energy consumption [107], a more holistic evaluation protocol would need to include the not insignificant manufacturing cost of the training infrastructure [136, 137]. The collaborative training method described in this work allows volunteers to make better use of existing computing resources, which helps minimize these costs. Second, the distributed training setting allows users to dispense with the extensive cooling infrastructures required for large concentrated data centers, and may thus also help reduce the operating costs themselves [138]. We note, however, that the additional networking needs may limit the magnitude of these gains.