# OpenReview forum: "Distributed Deep Learning In Open Collaborations"
_NeurIPS.cc/2021/Conference — NeurIPS 2021 Poster_

### Official Review · Reviewer_vAEP · 2021-07-05

**Rating:** 5
**Confidence:** 4

**Summary:**

This paper attempts to deploy distributed training of deep neural networks over an open collaborative distributed environment, where participators can dynamically join the computational task.
The approach is mainly based on modification of data parallel training to overcome the technique challenges introduced by the collaborative environment.
Both standard and real-world pre-train benchmarks are included to evaluate the proposed system.

**Ethical Concerns:**

No applicable.

**Limitations And Societal Impact:**

Yes.

**Main Review:**

This paper talks about a very interesting problem, where fragmented computational resource would be able to more efficiently leveraged for jointly training deep neural networks. As such models can only be trained inside data-center clusters currently, the approach proposed in this paper would be broadly beneficial to a very large research area.

However, the methodology proposed in this paper is relatively limited:
+ First, the method limited itself to data parallel training, however, this would introduce inconsistency with the motivation of the work. In a dynamic environment, it is unwanted to assume the whole model can be hold in a single participator's RAM, which is the fundamental constraint for data parallel training.
+ Further, even limiting the design to the data parallel approach, the approach solves the problem by simply extending very large batch training at the algorithmic level. However, very large batch training would compromise the generalization performance in some cases. On the other hand, there could be lots of other design choices neglected by the paper, for example, asynchronous SGD and local SGD are proposed to replace large batch training. At least some discussion to justify the proposed design choice would be necessary.

The paper is not well polished for its writing.
+ There are lots of typos throughout the paper: e.g., Line 204 "peer i peer"; Line 319 "Figure 1" (it should be Table 1 I conjecture).
+ Figure 2 is confusing, what do the rectangles and lines represent?

There are some flaws in the empirical study:
+ Since the benchmarks are based on some pre-training and fine-tuning tasks, it looks a little unconfident to just report the training loss. At least some results about the generalization performance should be reported. For example, for the SwAV task, at least some linear model should be used to justify the proposed approach can reach a similar level of accuracy reported in the original paper.

+ There is a missing of scalability benchmarks, for example, when adding more instances, how would the system scale out. It is at least reasonable to report  the training sample throughput when including 4, 8, 16, 32, etc instances for one particular setting.

+ The experimental section is not well presented, there is a missing of clear hypothesis of the experiments.

Post rebuttal update:

I really appreciate the great effort the author has made to address my concerns. Based on these feedbacks, I have some follow-up comments:
+ It seems that the proposed method can achieve an almost linear speedup w.r.t training sample throughput, which is awesome. On the other hand, I think there could be some analysis about why the proposed approach can reach this desired performance, i.e., given the model size and the network bandwidth, how the communication is hidden inside the computation slot (considering the FLOP in the forward and backward pass and FLOPs of the GPU devices.)
+ I believe the paper should clearly state the category of the training tasks the proposed work can be applied, e.g., fine-tuning tasks where large batch training can be adopted.
+ The writing can be further polished according to the helpful feedbacks from all the reviewers.

**Time Spent Reviewing:**

10

---

> ### Author Response · Authors · 2021-08-06
> **References for Author Response**
>
> [1]  Jie Ren, Samyam Rajbhandari, Reza Yazdani Aminabadi, Olatunji Ruwase, Shuangyan Yang, Minjia Zhang, Dong Li, and Yuxiong He. Zero-offload: Democratizing billion-scale model training, 2021.
>
> [2] Bharadwaj Pudipeddi, Maral Mesmakhosroshahi, Jinwen X, and Sujeeth Bharadwaj. Training Large Neural Networks with Constant Memory using a New Execution Algorithm, 2021
>
> [3] Yu Sun, Shuohuan Wang, Shikun Feng, Siyu Ding, Chao Pang, Junyuan Shang, Jiaxiang Liu, Xuyi Chen, Yanbin Zhao, Yuxiang Lu, Weixin Liu, Zhihua Wu, Weibao Gong, Jianzhong Liang, Zhizhou Shang, Peng Sun, Wei Liu, Xuan Ouyang, Dianhai Yu, Hao Tian, Hua Wu, and Haifeng Wang. ERNIE 3.0: Large-scale Knowledge Enhanced Pre-training for Language Understanding and Generation, 2021
>
> [4] Lan, Z.-Z., Chen, M., Goodman, S., Gimpel, K., Sharma, P., and Soricut, R. Albert: A lite bert for self-supervised learning of language representations. In International Conference on Learning Representations, 2020.
>
> [5] Yanping Huang, Youlong Cheng, Ankur Bapna, Orhan Firat, Dehao Chen, Mia Chen, HyoukJoong Lee, Jiquan Ngiam, Quoc V Le, Yonghui Wu, et al. Gpipe: Efficient training of giant neural networks using pipeline parallelism. In Advances in Neural Information Processing Systems, pages 103–112, 2019.
>
> [6] Deepak Narayanan, Mohammad Shoeybi, Jared Casper, Patrick LeGresley, Mostofa Patwary, Vijay Korthikanti, Dmitri Vainbrand, Prethvi Kashinkunti, Julie Bernauer, Bryan Catanzaro, et al. Efficient large-scale language model training on gpu clusters. arXiv preprint arXiv:2104.04473, 2021.
>
> [7]  Yang You, Igor Gitman, and Boris Ginsburg. Large batch training of convolutional networks, 2017.
>
> [8] Ting Chen, Simon Kornblith, Mohammad Norouzi and Geoffrey Hinton. A Simple Framework for Contrastive Learning of Visual Representations, ICML’2020.
>
> [9]  Mathilde Caron, Ishan Misra, Julien Mairal, Priya Goyal, Piotr Bojanowski, and Armand Joulin. Unsupervised learning of visual features by contrasting cluster assignments. In H. Larochelle, M. Ranzato, R. Hadsell, M. F. Balcan, and H. Lin, editors, Advances in Neural Information Processing Systems, volume 33, pages 9912–9924. Curran Associates, Inc., 2020.
>
> [10]  Yang You, Jing Li, Sashank Reddi, Jonathan Hseu, Sanjiv Kumar, Srinadh Bhojanapalli, Xiaodan Song, James Demmel, Kurt Keutzer, and Cho-Jui Hsieh. Large batch optimization for deep learning: Training bert in 76 minutes. In International Conference on Learning Representations, 2020.
>
> [11] Jared Kaplan, Sam McCandlish, Tom Henighan, Tom B. Brown, Benjamin Chess, Rewon Child, Scott Gray, Alec Radford, Jeffrey Wu, and Dario Amodei. Scaling laws for neural language models, 2020.
>
> [12] Martin Popel and Ondˇrej Bojar. Training tips for the transformer model. The Prague Bulletin of Mathematical Linguistics, 110, 03 2018.
>
> [13] Tom B Brown, Benjamin Mann, Nick Ryder, Melanie Subbiah, Jared Kaplan, Prafulla Dhariwal, Arvind Neelakantan, Pranav Shyam, Girish Sastry, Amanda Askell, et al. Language models are few-shot learners. arXiv preprint arXiv:2005.14165, 2020.
>
> [14] Zhewei Yao, Amir Gholami, Daiyaan Arfeen, Richard Liaw, Joseph Gonzalez, Kurt Keutzer and Michael Mahoney. Large batch size training of neural networks with adversarial training and second-order information, 2018

---

> ### Author Response · Authors · 2021-08-06
> **Author Response to Reviewer vAEP**
>
> We thank the reviewer for their insightful suggestions and comments. Below, we address each comment separately and announce additional experiments.
>
> > In a dynamic environment, it is unwanted to assume the whole model can be hold in a single participator's RAM, which is the fundamental constraint for data parallel training.
>
> This is indeed a shortcoming of DeDLOC and all traditional data-parallel methods. Fortunately, there is a recent line of work that alleviates this issue through the use of memory offloading. For instance, ZeRO-Offload [1] allows a user to train up to 4.5x larger models by offloading the optimizer and gradient accumulators to the CPU. Another study [2] offloads entire layers onto RAM, allowing them to train up to 50 billion parameter models on a single GPU with 16GB memory. While this is not the largest model ever trained, it is several times larger than the current SoTA in MLM Transformers [3] and approximately 200 times larger than the largest model from the ALBERT paper [4]. Thus, even without model parallelism, DeDLOC can be used to train the vast majority of deep learning models used in practice.
>
> Using model parallelism with volunteers is an interesting direction for future research. Without proper modification, model-parallel techniques[5,6] would irreversibly lose trainable parameters after any peer failure.
>
> > large batch training would compromise the generalization performance in some cases
>
> This is indeed true in some cases, especially when the training data is scarce and overfitting is a big concern. Fortunately, the computationally intensive tasks that make most sense for collaborative training typically use large datasets, where this problem is much less pronounced. Prior work demonstrates the efficiency of large batch training for both convolutional networks[7,8,9] and Transformers[10, 11]. Furthermore, that large-batch training can even improve the performance and stability in transformers[12, 13] and adversarial networks[14].
>
> > (Figure 2) what do the rectangles and lines represent?
>
> The rectangles represent different peers and the lines represent network bandwidth. For instance, in the middle section, the top-left node can communicate with other nodes faster than they can communicate with each other. The intention was to show how our averaging algorithm can adapt to different network conditions. We agree that the clarity of Figure 2 can be improved and will add a detailed legend in the final version of the paper.
>
> > it looks a little unconfident to just report the training loss
>
> We respectfully disagree with the premise. We report downstream evaluation metrics of sahajBERT in Table 2 and the full evaluation procedure in Appendix H.6. As for Section 4.1, we did not report the downstream quality, because we believed that the stepwise learning curves were too similar to be interesting (similarly to Figure 9). However, we agree that reporting the SwAV downstream performance will strengthen the work and will do so in the final version.
>
> > clear hypothesis of the experiments
>
> We thank the reviewer for this suggestion. To that end, we will modify the L296-301 to reflect that:
>
> Section 4.1 verifies if DeDLOC can indeed operate in our target setup with high performance;
>
> Section 4.2 studies how different collaboration types affect the training throughput;
>
> Section 4.3 reports the “view from trenches” of a real-world collaborative training experiment.
>
> > scalability benchmarks, 4, 8, 16, 32, etc instances for one particular setting.
>
> Since we target heterogeneous hardware, the training throughput of DeDLOC depends not only on the number of instances but also on their hardware composition. To that end, the experiments in Figure 4 and Table 1 are intended to show how DeDLOC’s throughput changes from different kinds of scaling, i.e. changing the number of regular workers, non-GPU peers, worker uptime, etc. That said, we concur with the reviewer and plan to add more conventional scalability experiments by the beginning of the discussion period (August 10).

---

### Official Review · Reviewer_HY7V · 2021-07-16

**Rating:** 5
**Confidence:** 4

**Summary:**

The paper discusses a distributed training setup where multiple small institutes/groups pool computational resources together for training ML models collaboratively. The paper focuses on two key problems: (1) maintaining consistent training outcomes under dynamic composition of participants. (2) determining the communication strategy adaptively based on dynamic participants & network conditions.

**Main Review:**

**Quality and Clarity**: Poor.

It appears that the paper tries to touch too many problems (optimization, communication, system design, self-supervised learning) at once while did not discuss any topic in depth. It is unclear what its core technical contributions are. For example, the paper discussed training consistency but only adopted synchronous data-parallel training and extremely large batches without showing how the consistency problem is resolved from optimization perspective. The paper also discussed adaptive communication strategies: all reduce, parameter servers, and hybrid while in fact these are not mutually exclusive (one can implement all reduce with parameter servers).

**Weaknesses**

1. Re section "3.1 ensuring training consistency", the authors argue that multiple runs under dynamic participants can lead to different results and claim synchronous data-parallel training with fixed hyperparameters and large batches can resolve this issue. However this is little analysis supporting this claim. Even the authors state in L164 that "synchronous updates makes DeDLOC mathematically equivalent to large-batch training on a regular cluster", it does not resolve the issue that the training would converge to different solutions with dynamic participants.

2. The authors did not discuss how data are distributed on different participants. The system heterogeneity issue are different from the data heterogeneity issue. It looks like this paper focuses on the former by discussing hardware/network/bandwidth differences. This leads to two concerns: (1) there is little insights presented in this paper as the staleness, stragglers, unstable hardwares are well studied for distributed training in computer clusters. And in fact the authors did not offer any discussion or comparison on this. (2) without discussing the data heterogeneity, the notion of "collaborative learning" is overly broad and therefore misleading.

3. The experiment setup is flawed. In L297, the authors say that their primary focus is "training models that are useful for a wide range of downstream tasks" and then show results from two self-supervised learning experiments. The authors should have compare the proposed method with alternatives algorithms or systems instead.

4. The paper's title is "Distributed Deep Learning in Open Collaboration". It's not clear which part of the paper is unique to deep learning.

**Originality** and **Significance**: limited due to the concerns above.

The paper could be stronger if the authors focus on one core technical challenges, analysis it in depth, and compare the proposed method with alternative methods to show its superiority.

=====

**Post rebuttal comments**

Thanks to the authors for the thorough response. I have read the response and all other reviews too. I agree the paper addresses an interesting problem in heterogeneous training and provides a linear programming based adaptive averaging algorithm for computing optimal strategies. Despite the good empirical results and thorough presentation on system design, a main shortcoming of the paper is the lack of rigorous analysis from the optimization perspective. With training consistency being one of the main challenges, the technical proposal presented in section 3.1 and 3.3 as well as in the rebuttal are rather handwavy -- highly empirical and procedural in system design. Overall, I am raising my rating to 5: Marginally below the acceptance threshold.

**Time Spent Reviewing:**

4

---

> ### Author Response · Authors · 2021-08-04
> **References for Author Response**
>
> [1] DeBERTa: Decoding-enhanced BERT with Disentangled Attention. Pengcheng He, Xiaodong Liu, Jianfeng Gao, Weizhu Chen. arXiv:2006.03654
>
> [2] Emerging Properties in Self-Supervised Vision Transformers. Mathilde Caron, Hugo Touvron, Ishan Misra, Hervé Jégou, Julien Mairal, Piotr Bojanowski, Armand Joulin.      arXiv:2104.14294
>
> [3] DynaSent: A Dynamic Benchmark for Sentiment Analysis. Christopher Potts, Zhengxuan Wu, Atticus Geiger, Douwe Kiela. ACL 2021
> [4] A Broad-Coverage Challenge Corpus for Sentence Understanding through Inference. Adina Williams, Nikita Nangia, Samuel R. Bowman. NAACL 2018
>
> [5] Know What You Don't Know: Unanswerable Questions for SQuAD. Pranav Rajpurkar, Robin Jia, Percy Liang. ACL 2018
>
> [6] SuperGLUE: A Stickier Benchmark for General-Purpose Language Understanding Systems. Alex Wang, Yada Pruksachatkun, Nikita Nangia, Amanpreet Singh, Julian Michael, Felix Hill, Omer Levy, Samuel R. Bowman. NeurIPS 2019
>
> [7] XTREME: A Massively Multilingual Multi-task Benchmark for Evaluating Cross-lingual Generalization. Junjie Hu, Sebastian Ruder, Aditya Siddhant, Graham Neubig, Orhan Firat, Melvin Johnson. ICML 2020.
>
> [8] BERT: A Review of Applications in Natural Language Processing and Understanding. M. V. Koroteev. arXiv:2103.11943
>
> [9] Unsupervised Learning of Visual Features by Contrasting Cluster Assignments. Mathilde Caron, Ishan Misra, Julien Mairal, Priya Goyal, Piotr Bojanowski, Armand Joulin. NeurIPS 2020
>
> [10] A Unified Architecture for Accelerating Distributed DNN Training in Heterogeneous GPU/CPU Clusters. Yimin Jiang, Yibo Zhu, Chang Lan, Bairen Yi, Yong Cui, and Chuanxiong Guo. OSDI 2020
>
> [11] Pytorch Distributed: Experiences on Accelerating Data Parallel Training. Shen Li, Yanli Zhao, Rohan Varma, Omkar Salpekar, Pieter Noordhuis, Teng Li, Adam Paszke, Jeff Smith, Brian Vaughan, Pritam Damania, and Soumith Chintala. VLDB 2020
>
> [12] Horovod: fast and easy distributed deep learning in Tensorflow. Sergeev, Alexander & Balso, Mike. arXiv:1802.05799
>
> [13] A Survey on Distributed Machine Learning. Joost Verbraeken, Matthijs Wolting, Jonathan Katzy, Jeroen Kloppenburg, Tim Verbelen, and Jan S. Rellermeyer. ACM Computing Surveys, Volume 53, Issue 2, 2020
>
> [14] Communication-Efficient Distributed Deep Learning: A Comprehensive Survey. Zhenheng Tang, Shaohuai Shi, Xiaowen Chu, Wei Wang, Bo Li. arXiv:2003.06307
>
> [15] Bandwidth optimal all-reduce algorithms for clusters of workstations. Pitch Patarasuk and Xin Yuan. Journal of Parallel and Distributed Computing, 2009.

---

> ### Author Response · Authors · 2021-08-04
> **Author Response to Reviewer HY7V**
>
> We thank the reviewer for their feedback. Below we address the raised concerns to the best of our ability.
>
> > Authors say that their primary focus is "training models that are useful for a wide range of downstream tasks" and then show results from two self-supervised learning experiments
>
> As we state in L296-299, both chosen models are useful for a wide range of downstream tasks, and transfer learning from universal representations obtained by self-supervised learning achieves highly competitive results in different applications[1,2]. For instance, pretrained language models are used for sentiment analysis[3], natural language inference[4], question answering[5], and numerous other tasks [6,7,8]. In turn, SwAV can be used for tasks like few-shot image classification, object detection and instance segmentation, as discussed in the original paper [9].
>
> > Authors should have compared the proposed method with alternatives algorithms or systems instead.
>
> In Table 1, we compare our adaptive averaging algorithm with standard parameter server and all-reduce in terms of training throughput (since all algorithms produce the same results with different efficiency). In Figure 6, we compare training efficiency of DeDLOC with volunteer devices against AR-SGD on traditional hardware. To account for the hardware imbalance, we also compare per-iteration convergence in Appendix H.5.
>
> > Authors did not discuss how data are distributed on different participants.
>
> We discuss the general approach for training on large datasets L280-291 and provide details in Appendix H.3.
>
> > The system heterogeneity issue is different from the data heterogeneity issue.
>
> We agree that training with heterogeneous data is a very important problem. However, in this study we focus on training on **public** datasets and use collaborative training to address the compute intensity. In contrast, scenarios involving heterogeneous data typically arise in Federated Learning. We provide a more detailed comparison between Federated Learning and our setup in Appendix A (referenced in L107-111).
>
> > How is the consistency problem resolved from the optimization perspective?
>
> We thank the reviewer for highlighting this question and plan to include a more detailed description of this issue in Section 3.3 and the corresponding appendix.
>
> DeDLOC ensures that all peers use up-to-date parameters (and send up-to-date gradients) by tracking the number of global steps attended by each peer. If, for instance, a peer temporarily hangs and skips a step, it will observe that other peers made more steps and will consider itself out-of-date. An out-of-date peer must immediately download the latest model parameters and optimizer statistics from one of the up-to-date peers before it is allowed to train again. Similarly, if a peer joins midway through training, it will download this “snapshot” from one of the up-to-date peers.
>
> > [The algorithm] does not resolve the issue that the training would converge to different solutions with dynamic participants
>
> This is technically correct: running DeDLOC with dynamic peers would be equivalent to randomly selecting minibatches with different seeds from the common dataset. However, deep learning models are robust to this inherent stochasticity. We validate this empirically by comparing DeDLOC against conventional AR-SGD in Appendix H.5.
>
> > Supporting the claim that DeDLOC is equivalent to AR-SGD in terms of iterations.
>
> Informally, all peers train on i.i.d. training examples and the optimizer step triggers when peers accumulate a predefined target batch size, which is exactly the same as in AR-SGD with the exception that batch size could be slightly larger.
> However, we agree that this argument is better supported with a formal analysis. We will provide such an analysis by the beginning of the discussion period (August 10).
>
> > All-reduce and Parameter Server are not mutually exclusive. One can implement all-reduce with parameter servers.
>
> All-reduce and parameter server are widely considered to be two distinct *system designs* in distributed training literature[10,11,12,13,14]. Technically speaking, one can indeed implement the all-reduce *collective operation* by sending all data through a central “parameter server”. However, this would be less efficient and significantly less scalable than standard all-reduce protocols[15], which is significant in our communicationally constrained setting.
>
> > The paper's title is "Distributed Deep Learning in Open Collaboration". It's not clear which part of the paper is unique to deep learning.
>
> DeDLOC is tailored to the typical requirements of deep learning, such as stochastic optimization over millions of trainable parameters (S3.1) or working with large training datasets (L260-291). Some of the proposed ideas may find other practical applications; however, each of these alternatives deserves a separate investigation that would not fit within the scope of one paper.

---

### Official Review · Reviewer_d2iV · 2021-07-16

**Rating:** 8
**Confidence:** 4

**Summary:**

The paper proposes *DeDLOC* framework to enable distributed Deep Learning in open collaboration by leveraging the computing resources in a volunteering user setup. The framework adapts to the available hardware and communication network to maximize training throughput and accordingly behaves as PS  [33], AR-SGD [34], decentralized SGD [35], BytePS [36], or a combination of them in hybrid mode. The experiments are done with real training collaboration across 49 volunteers for Bengali language modeling using ALBERT-large. This is a first-of-its-kind volunteering effort for collaborating training. Additionally, the proposed framework is also applied to self-supervised tasks under 3 compute setups (WORKSTATION: homogeneous low bandwidth, SERVER: homogeneous high bandwidth, and HYBRID: heterogeneous)

**Ethical Concerns:**

The authors have discussed this

**Limitations And Societal Impact:**

The authors have discussed it sufficiently

**Main Review:**

**Originality:**

The work is nicely motivated and provides a practical solution validated with real experiments. The problem setup is novel and clearly addresses key challenges in open collaboration environments (Sec 3.1 to Sec 3.3). The paper clearly distinguishes itself from related work and provides an adaptive averaging framework that can be modeled as one of the various popular data-parallel distributed deep learning strategies (PS, AR-SGD, etc) based on collaboration setups to compute the optimal solution.


**Quality:**

The paper provides a linear programming optimization problem to model optimal communication strategy with the highest training throughput while the participants are joining and leaving. The work is complete in its dealing with the challenges posed and presenting a practical solution to democratize deep learning and provide opportunities for unconventional datasets.


**Clarity:**

The paper is very well-written and provides a great narrative for the reader to think about the current challenges in real-world deployment and applications of distributed deep learning. The work is a bold attempt to inspire future research in open collaborative systems.

Few typos
- Line 98:  ... them ~are~ highly...
- Line 132: Space between period (.) and *"This specific problem..."*


**Significance:**

This is important work and the authors formulate the problem carefully to motivate the community to focus on large-scale collaborative distributed training strategies. The experiments are the highlight of the work and the results are a stepping stone to incorporate volunteer setups.

**Time Spent Reviewing:**

2

---

> ### Author Response · Authors · 2021-08-06
> **Author Response to Reviewer d2iV**
>
> We thank the reviewer for their encouraging feedback and a detailed summary of our contributions. We will fix the typos in the final version of the paper.

---

### Official Review · Reviewer_Z6aL · 2021-07-18

**Rating:** 7
**Confidence:** 4

**Summary:**

The authors studied an important topic for many potential future applications in this paper.

Collaborative training means different users in different physical locations share their computing resources to train a deep learning model together.

Otherwise, a single user does not have enough computing resources to finish the training.

Overall, this paper is well written. This could be an important solution for the unbalanced geographic computing resources distribution.

**Main Review:**

The authors studied an important topic for many potential future applications in this paper.

Collaborative training means different users in different physical locations share their computing resources to train a deep learning model together.

Otherwise, a single user does not have enough computing resources to finish the training.

The focus of this paper is to analyze these constraints and propose a novel algorithmic framework designed specifically for collaborative training.

The authors provided some strong empirical results based on realistic workloads such as SwAV and ALBERT pretraining to support the proposed method.

Overall, this paper is well written. This could be an important solution for the unbalanced geographic computing resources distribution.


We know the size of the communicated gradients is equal to the size of the model.

Therefore, communication overhead could become a big concern to super-large models like GPT-3 (1750 billion parameters) in future real-world deep learning applications.

One feature of DeDLOC is to reduce the communication volume by large-batch optimization to speed up the training process, which seems to be very practical.

Another feature is the adaptive averaging algorithm, which essentially makes the users can get the optimal solution in different situations. Figure 1 and Figure 2 provided some good illustrations.


Another contribution for this work is that their efforts for implementation. I believe the engineering part of this work is non-trivial.


Some of my questions for future work:

Is it a better way of collaboration for different people to share their quota on cloud and create a large-scale stable computing pool?

How can people quantize their contributions and benefits in open computing collaboration? Some users may contribute less computing resources, but their computing power is very high, which may be more useful.

Did the authors tried conducting a convergence analysis for the proposed method?

**Time Spent Reviewing:**

6 hours

---

> ### Author Response · Authors · 2021-08-04
> **Author Response to Reviewer Z6aL**
>
> We thank the reviewer for their feedback, encouragement, and insightful questions.
>
> >How can people quantize their contributions and benefits in open computing collaboration?
>
> This is indeed an important aspect. In the collaborative experiment from Section 4.3, peers kept track of their “lifetime contribution” by counting the total number of samples for which they accumulated gradients. As such, if peer A has a more powerful GPU than peer B, it will be able to accumulate “contributions” faster. This “contribution score” proved to be an effective way to engage volunteers, who would sometimes compete with each other on the public contribution leaderboard.
>
> However, this heuristic score proved has some caveats, e.g. it does not reward peers for their role in the gradient aggregation phase. As such, we will add a broader discussion of this issue in Appendix H.
>
> >Is it a better way of collaboration for different people to share their quota on the cloud and create a large-scale stable computing pool?
>
> There are two main scenarios that can arise. If all participants have access to homogeneous hardware in the same region and cloud provider, they can allocate preemptible GPU instances and train using “elastic” distributed training frameworks: PyTorch Elastic, Elastic Horovod, or DeDLOC.
>
> If any of these conditions are not met, (for instance, quotas are with different cloud providers), DeDLOC provides a way to pool together different cloud instances with uneven GPU compute, bandwidth and reliability.
>
> >Convergence analysis for the proposed method
>
> While we make several informal claims about convergence, the paper would indeed benefit from a formal convergence analysis. In addition, Reviewer HY7V also requested that we formally prove the relation between DeDLOC and conventional synchronous SGD.
>
> We will provide this analysis in a common response before the start of the discussion period (August 10).

---

### Author Response · Authors · 2021-08-10
**Common response**

We thank the reviewers for their valuable feedback. All reviewers except HY7V agree that this paper touches on an "important topic for many potential future applications" (Z6aL) and "would be broadly beneficial to a very large research area" (vAEP). We did our best to address the reviewers' questions in the individual review threads and will eagerly continue the discussion. Below, we answer some of the common questions and report additional experiments, as requested.

**(vAEP) Scalability benchmarks, 4, 8, 16, 32, etc instances for one particular setting**

As promised, we ran scalability benchmarks for DeDLOC using the same baseline setup from Section 4.2 (ALBERT). Below we report the time (in seconds) per SGD step for different collaboration scales.

| # nodes  |  8 nodes   | 16 nodes   | 32 nodes | 64 nodes |
| ------------ | ------------ | ---------------| ------------ | ------------- |
| sec./step | 56.38576 | 28.563636 | 16.89247 | 9.62576  |


**(vAEP) Additional downstream evaluation for SwAV**

The longest-training experiment in that section was a hybrid setup, that observed approximately 200M training samples, which translates to ≈153 epochs. We evaluated this checkpoint using the same protocol as described in Section 4.1 from the SwAV paper [1] and obtained the validation accuracy of 0.72176. This agrees with the original training results in Figure 3 of the SwAV paper.

**(Z6aL, HY7V) Convergence analysis**

DeDLOC updates parameters only after accumulating the target number of gradients from up-to-date peers. However, peers can accumulate more samples due to network delays. Thus, we can analyze DeDLOC as a regular SGD with varying batch sizes. This allows us to reuse some of the existing convergence bounds from the optimization literature.

More formally, consider a standard optimization problem $\min_{x \in \mathbb{R}^n} f(x)$, solved by SGD. We denote the gradients for step k as $\operatorname{E}[g^k|x^k]=\nabla f(x^k)$  and the corresponding update as $x^{k+1} = x^k - \gamma_k g^k$.

Let’s denote the variance of a single stochastic gradient as $\operatorname{E}[\|\nabla f(x^k, \xi^k_i)-\nabla f(x^k)\|^2|x^k]\leq\sigma^2_0$ and the target batch size as $m$ .

At step $k$, DeDLOC will accumulate gradients from $m_k \ge m$ individual samples: $$g^k= \frac{1}{m_k}\sum^{m_k}_{i = 1}\nabla f(x^k, \xi^k_i).$$

Thus, the gradient averaged over a minibatch of i.i.d. $m_k$ samples will have the variance:
$$ \operatorname{E}[\|g^k-\nabla f(x^k)\|^2|x^k] = \frac{1}{m^2_k}  \sum_{i = 1}^{m_k} \operatorname{E}[\|\nabla f(x^k, \xi^k_i)-\nabla f(x^k)\|^2|x^k] \leq \frac{1}{m^2_k}\sum^{m_k}_{i = 1} \sigma^2_0.$$

Since $m_k \ge m$,

$$\frac{1}{m^2_k}\sum^{m_k}_{i = 1} \sigma^2_0 = \frac{\sigma^2_0 }{m_k} \leq \frac{\sigma^2_0 }{m} = \sigma^2$$

This allows us to reuse the existing SGD convergence bounds from the optimization literature, e.g. [2] or [3]. For instance, we can use Theorem 5 from [2] and plug in $\sigma^2_0 / m$ as the gradient variance and get

$$ \operatorname{E}{ f(\bar x_T) - f^\star} + \mu \operatorname{E}{\|x_{T+1}-x^\star}\|^2 \leq \min \left \\{ 64 L R^2 \exp \left[-\frac{\mu T}{4L} \right] + \frac{36 \sigma_0^2}{\mu m T} ,  \frac{2LR^2}{T} + \frac{2 \sigma_0 R}{\sqrt{mT}}  \right \\}.$$


(we are using the notation from [2] in this section)


**References**


[1] Unsupervised Learning of Visual Features by Contrasting Cluster Assignments. Mathilde Caron, Ishan Misra, Julien Mairal, Priya Goyal, Piotr Bojanowski, Armand Joulin. NeurIPS 2020

[2] Unified Optimal Analysis of the (Stochastic) Gradient Method, Sebastian U. Stich, 2019

[3] Ahmed Khaled, Othmane Sebbouh, Nicolas Loizou, Robert M. Gower, Peter Richtárik, Unified Analysis of Stochastic Gradient Methods for Composite Convex and Smooth Optimization, 2020

---

### Decision · Program_Chairs · 2021-09-27

**Decision:**

Accept (Poster)

**Comment:**

Although there was a spread of opinions from the reviewers, they did all agree that the problem area addressed by the paper is both interesting and important.  Indeed, finding ways to create more opportunities for those outside of select industry or academic groups to participate in massive scale model training is important from both a research perspective and from a community perspective that takes into account the overall health of the field into account.

Additionally, while the reviewers initially raised some important concerns, these have been well addressed by the author responses.  In particular, the scaling results and the convergence analysis are both very helpful, and should be included in the final version in some way.

One thing to note is that the spirit of reviewer HY7V's comment that the paper is trying to address many (important) issues is indeed a factor here, as evidenced by the repeated references to the (extensive) appendix.  I recognize that the page limits of NeurIPS are a significant constraint, and that the "best" form of this paper is likely a journal article that incorporates the appendix information more fully into the main text and narrative.  That said, taking all of the reviews into account and the ensuing discussion, I do think that this conference-paper version has significant merit and will spark interest and discussion within the field.